



# European primary emissions of criteria pollutants and greenhouse gases in 2020 modulated by the COVID-19 pandemic disruptions

Marc Guevara[1], Hervé Petetin[1], Oriol Jorba[1], Hugo Denier van der Gon[2], Jeroen Kuenen[2], Ingrid Super[2], Jukka-Pekka Jalkanen[3], Elisa Majamäki[3], Lasse Johansson[3], Vincent-Henri Peuch[4] and Carlos Pérez Garcia-Pando[1,5]

[1] Barcelona Supercomputing Center, Barcelona, Spain
[2] TNO, Department of Climate, Air and Sustainability, Utrecht, the Netherland
[3] Atmospheric Composition Research, Finnish Meteorological Institute, 00560 Helsinki, Finland
[4] European Centre for Medium-Range Weather Forecasts, Reading, UK
[5] ICREA, Catalan Institution for Research and Advanced Studies, Barcelona, Spain

*Correspondence to*: Marc Guevara (marc.guevara@bsc.es)

**Abstract.** We present a European dataset of daily-, sector-, pollutant- and country-dependent emission adjustment factors associated to the COVID-19 mobility restrictions for the year 2020. The resulting dataset covers a total of nine emission sectors, including road transport, energy industry, manufacturing industry, residential and commercial combustion, aviation, shipping, off-road transport, use of solvents, and fugitive emissions from transportation and distribution of fossil fuels. The dataset was produced to be combined with the Copernicus CAMS-REG_v5.1 2020 business-as-usual (BAU) inventory, which provides high resolution (0.1 x 0.05 deg.) emission estimates for 2020 omitting the impact of the COVID-19 restrictions. The combination of both datasets allows quantifying spatially- and temporally-resolved reductions in primary emissions from both criteria pollutants ($NO_x$, $SO_2$, NMVOC, $NH_3$, CO, $PM_{10}$ and $PM_{2.5}$) and greenhouse gases ($CO_2$ fossil fuel, $CO_2$ biofuel and $CH_4$), as well as assessing the contribution of each emission sector and European country to the overall emission changes. Estimated overall emission changes in 2020 relative to BAU emissions were as follows: -10.5% for $NO_x$ (-602 kt), -7.8% (-260.2 Mt) for $CO_2$ from fossil fuels, -4.7% (-808.5 kt) for CO, -4.6% (-80kt) for $SO_2$, -3.3% (-19.1 Mt) for $CO_2$ from biofuels, -3.0% (-56.3 kt) for $PM_{10}$, -2.5% (-173.3 kt) for NMVOC, -2.1% (-24.3 kt) for $PM_{2.5}$, -0.9% (-156.1 kt) for $CH_4$ and -0.2% (-8.6 kt) for $NH_3$. The most pronounced drop in emissions occurred in April (up to -32.8% on average for $NO_x$) when mobility restrictions were at their maxima. The emission reductions during the second epidemic wave between October and December, were three to four times



lower than those occurred during the Spring lockdown, as mobility restrictions were generally softer (e.g., curfews, limited social gatherings). Italy, France, Spain, the United Kingdom and Germany were, together, the largest contributors to the total EU27 + UK absolute emission decreases. At the sectoral level, the largest emission declines were found for aviation (-51 to -56%), followed by road transport (-15.5% to -18.8%), the latter being the main driver of the estimated reductions for the majority of

pollutants. The collection of COVID-19 emission adjustment factors (https://doi.org/10.24380/k966-3957, Guevara et al., 2022) and the CAMS-REG_v5.1 2020 BAU gridded inventory (https://doi.org/10.24380/eptm-kn40, Kuenen et al., 2022) have been produced in support of air quality modelling studies.

## 1    Introduction

The COVID-19 pandemic lockdowns and mobility restrictions implemented across Europe have resulted in an unprecedented drop in atmospheric anthropogenic emissions. Using satellite and in situ observations, several studies have reported the associated changes in air pollutants (e.g., Balamuguran et al., 2021; Barré et al., 2021; Grange et al., 2021; Petetin et al., 2020; Querol et al., 2021; Slezakova and Pereira, 2021), mostly focusing on main criteria pollutants (i.e., mostly $NO_2$ and $O_3$, as well as PM10 and

$PM_{2.5}$ to a lesser extent) during the so-called Spring lockdowns and the immediate period thereafter (i.e., between mid-March and July). Further insights that complement these observational studies can be obtained by quantifying the changes in primary emissions. Such quantification can unlock many possibilities for numerical modelling studies, which require gridded emissions that account for the effect of the pandemic. Also, understanding to what extent individual pollutant sources were affected along with

their associated emissions can provide valuable information to policy makers for the development of future abatement strategies.

Up to now, the number of studies tackling the impact of COVID-19 upon primary emissions is low compared to those focusing on air quality. At the global scale, Le Quéré et al. (2020 and 2021), Liu et al.

(2020a), Forster et al. (2020) and Doumbia et al. (2021) stand out. The first two focus on estimating the impact of the lockdowns on $CO_2$ emissions, while the other two quantify emission declines for both



criteria pollutants (NOx, SOx, NMVOCs, $NH_3$, $PM_{10}$ and $PM_{2.5}$) and greenhouse gases ($CO_2$ and $CH_4$). In all cases, results are reported at the daily, country and pollutant sector level. The estimates provided in Liu et al. (2020a) are continuously updated using near real time information provided by the Carbon

Monitor system (Liu et al., 2020b). In contrast, the datasets reported in Forster et al. (2020), Le Quéré et al. (2021) and Doumbia et al. (2021) focus on year 2020.

A common limitation in all the aforementioned works is related to the representativeness of certain datasets used to estimate changes in emissions. For instance, Forster et al. (2020) and Doumbia et al.

(2021) estimated emission changes for several sectors (i.e., road transport, residential/commercial combustion, manufacturing industry) relying on the trends reported by the Google COVID-19 Community Mobility Reports (Google LLC, 2021). However, the significant deviations between these new mobility datasets and traditional proxies such as traffic counts or energy consumption statistics, suggest caution in their use to assess emission changes (e.g., Harkins et al., 2020; Gensheimer et al.,

2021). In Liu et al. (2020a), changes in road transport emissions are based on changes in congestion levels reported by TomTom in 416 global cities in 57 countries. Since congestion levels do not directly reflect changes in the number of circulating vehicles, Liu et al. (2020) used a sigmoid function to fit a relationship between TomTom congestion levels and traffic counts, using as a proxy real measured traffic counts obtained for the city of Paris. The relationship found for Paris city was then applied to the TomTom

congestion levels reported for all other cities.

At the European scale, specific COVID-19 emission datasets have been developed mainly to perform air quality modelling studies. Menut et al. (2020) developed an emission scenario for western Europe that was limited to March 2020 and was set up using the Apple movement trends (Apple, 2021) to derive

emission reductions for road transport, manufacturing industry, non-road transport and residential–commercial combustion activities. Guevara et al. (2021) constructed a set of EU27 + UK daily COVID-19 emission adjustment factors for the most severe lockdown period (i.e., 21 February until 26 April 2020) and the sectors suffering the largest reductions in their activity: energy and manufacturing industry, road transport and aviation. In Matthias et al. (2021), the COVID-19 emission scenario was constructed



for central Europe and a total of five sectors (i.e., public power, manufacturing industry, road transport, shipping and aviation) and for the months of January to June 2020. Other sources of information besides mobility reports were used in Guevara et al. (2021) and Matthias et al. (2021), such as airport traffic statistics, electricity demand statistics or volume indexes of industrial production, among others. Of all the aforementioned works, only Guevara et al. (2021) reported its final emission dataset in open access.


This work represents an extensive update and refinement of the effort initially described in Guevara et al. (2021), including: (i) an extension of the temporal coverage to estimate the overall impact of the COVID-19 restrictions on the 2020 European emissions, (ii) the inclusion of anthropogenic sources previously not considered and (iii) the consideration of pollutant-dependent emission adjustment factors for both

criteria pollutants ($NO_x$, NMVOC, CO, $SO_2$, $NH_3$, $PM_{10}$, $PM_{2.5}$) and greenhouse gases ($CO_2$ from fossil fuel, later referred to as $CO_2\_ff$, $CO_2$ biofuel, later referred to as $CO_2\_bf$ and $CH_4$). As a result, we present an open-source dataset of European COVID-19 emission adjustment factors for the year 2020 that vary per day of the year, country (or sea region), sector and pollutant. The final set of adjustment factors covers the period from 21 February 2020, the beginning of localized lockdown in Italy (region of Lombardy), to

31 December 2020 and the following anthropogenic sources: public energy and heat production industry, manufacturing industry, residential and commercial combustion activities, use of solvents, fugitive emissions from production and transportation of fossil fuels, road transport, shipping, aviation (landing and take-off cycles) and other off-road transport sources. Adjustment factors were calculated using a wide range of open-access and near-real-time national measured activity data that resemble the effects of

lockdown measures on emissions released from multiple sources. This includes the combination of traditional proxies with new mobility metrics, meteorological parameters and machine learning techniques, among others.

The dataset is designed to reflect the heterogeneous impact of the lockdowns and mobility restrictions

across European countries and sectors, and to support the quantification of European primary emission changes. Accordingly, the emission adjustment factors were produced in a format consistent with the CAMS-REG gridded emission inventory (Kuenen et al., 2021a and b), developed under the Copernicus



Atmosphere Monitoring Service (CAMS) in direct support of the European regional production chain (Marécal et al., 2015). The annual emissions reported by CAMS-REG v5.1 for 2018 were extrapolated per country, sector and pollutant to 2020 neglecting the impact of COVID-19 to produce a business-as-usual (BAU) scenario. The combination of both datasets allows to spatially and temporally quantify reductions in primary emissions linked to the COVID-19 restrictions, as well as to assess the contribution of each pollutant sector to the overall emission changes.

The paper is organized as follows. Section 2 describes, for each sector, the approaches and sources of information used to construct the COVID-19 emission adjustment factors along with the resulting dataset. Section 3 presents the methodology used to produce BAU emissions for 2020. Section 4 compares the BAU and the COVID-19 emission scenarios. Section 5 provides a description of the data availability, and finally Sect. 6 presents the main conclusions of this work.

## 2    COVID-19 emission adjustment factors

The construction of the COVID-19 emission adjustment factors followed a data-driven approach. Changes in emissions are assumed to follow changes detected in measured time-series that represent the main activities of each pollutant sector at country level. For each sector, emission adjustment factors were calculated as a ratio between the activity data for a given day/week/month and the value of this activity over a pre-lockdown period (hereafter referred to as baseline).

The resulting dataset of adjustment factors follows the sector classification reported by the CAMS-REG-AP/GHG emission inventory, which corresponds to the Gridded aggregated Nomenclature For Reporting (GNFR). We considered twelve GNFR sectors, corresponding to nine pollutant sectors with road transport emissions split into 4 fuel types: GNFR_A (energy industry), GNFR_B (manufacturing industry), GNFR_C (other stationary combustion activities), GNFR_D (fugitive emissions from fossil fuel production and transportation), GNFR_E (solvents), GNFR_F1 (road transport: gasoline exhaust), GNFR_F2 (road transport: diesel exhaust), GNFR_F3 (road transport: LPG exhaust), GNFR_F4 (road transport: non-exhaust), GNFR_G (shipping), GNFR_ H (aviation) and GNFR_I (off road transport).






The time span of the adjustment factors of the current dataset is from 21 February to 31 December 2020. The beginning of the period corresponds to the date of the first localized lockdown in the region of Lombardy, Italy. The dataset covers: (i) the European first round of lockdowns, when mobility restrictions were at their maximum and remained almost unchanged for five weeks (mid-March until end of April),

(ii) the transition period towards the post-lockdown conditions (beginning of May until end of September), when national governments rolled-back COVID-19 measures, and (iii) the new round of lockdowns associated to the second pandemic wave in Europe (beginning of October until end of December), which forced governments back into mobility restrictions. In terms of spatial coverage, we included as many countries as possible that are covered by the CAMS-REG European working domain

(30° W – 60° E and 30° N – 72°N), giving priority to EU27 + UK, Norway and Switzerland. The spatial coverage of the adjustment factors constructed for each GNFR sector as well as a complete list of the countries considered is available in the Supplementary material (Table S1 and Figure S1).

Table 1 summarizes the main sources of information used to compute the adjustment factors for each

GNFR sector. For the GNFR_B, GNFR_C, GNFR_D, GNFR_E, GNFR_F2, GNFR_F4 and GNFR_I categories, subsector adjustment factors were first computed to take into account the heterogenous impact of the COVID-19 restrictions across the different emission sources in some sectors (e.g., light duty vehicles versus heavy duty vehicles in GNFR_F2 and GNFR_F4). The lists of subsectors considered for each GNFR category are listed in Table 2. The adjustment factors computed for each subsector were later

aggregated to the GNFR sector level by considering the relative contribution of each subcategory to total GNFR emissions, as expressed by Eq. (1):

$$EAF_{GNFR}(d, c, p) = \sum_1^N AF_{GN}(d, c) * S_{GN}(c, p) \tag{1}$$

where $EAF_{GNFR}(d, c, p)$ is the final emission adjustment factor for a given GNFR sector, for day $d$, country $c$ and pollutant $p$ [%]; $AF_{GN}(d, c)$ is the daily adjustment factor constructed for the subcategory $N$ of a given GNFR sector, for day $d$ and country $c$ [%] and $S_{GN}(c, p)$ is the contribution of the GNFR

subcategory $N$ to total GNFR emissions for country $c$ and pollutant $p$; being $N$ the total number of subcategories considered for a given GNFR sector (e.g. 3 for GNFR_B, 4 for GNFR_C, according to
Table 2).

As a result, pollutant-dependent adjustment factors were obtained for these seven GNFR sectors. The emission contributions from each subcategory to total GNFR emissions per country and pollutant (i.e., $S_{G01}(c, p)$, $S_{G02}(c, p)$) were computed using emissions from the GNFR_B, GNFR_C, GNFR_D,
GNFR_E, GNFR_F2, GNFR_F4 and GNFR_I sectors split following the subcategories listed in Table 2.

Figure 1 shows the resulting emission adjustment factors obtained per day, GNFR sector and selected pollutants. For all sectors except shipping, we show for illustrative purposes results for 6 European countries with different lockdown patterns (i.e., Italy, Spain, France, Germany, the United Kingdom and
Sweden). Italy was the country where restrictions first started, followed by Spain and France, where national lockdowns were imposed on 14 and 17 March, respectively. In contrast to Italy, where the transition from low to high stringency levels was gradual, these two countries experienced abruptly severe restrictions on movements, and commercial and industrial activities. A similar pattern occurred in Germany and the United Kingdom, where national lockdowns were imposed on the 20 and 23 March,
respectively. Sweden, on the other hand, was one of the few European countries where no national lockdown was implemented and only national recommendations (e.g., relatively soft social distancing measures) were provided to citizens.

The following subsections describe the data and methods for each sector along with the underlying
assumptions. The resulting adjustment factors reported in Fig.1 are also discussed in the corresponding subsection.

### 2.1.1   Public power industry

Changes in emissions from the public power sector (GNFR_A) were assumed to follow the changes observed in the electricity demand data reported by the European Network of Transmission System





Operators for Electricity (ENTSO-E) transparency platform (Hirth et al., 2018; ENTSO-E, 2021). For each country, we collected daily electricity demand data from January 2015 to December 2020. For Russia, Ukraine and Turkey we derived the electricity demand data from the corresponding national Transmission System Operators: SO-UPS (2021), UNEC (2021), and TEIAS (2021), respectively.

We first estimated the demand that would have occurred in the absence of COVID-19 under the same meteorological conditions, hereafter referred to as BAU. To estimate the BAU electricity demand we used gradient boosting machine (GBM) models trained and tuned independently for each country using daily data from January 2015 to December 2019. As inputs, we considered the following features: country-level daily population-weighted temperature ($T\_pop(d)$), date index (number of days since 2015/01/01),
Julian date, day of week and a Boolean feature indicating the country-specific bank holidays. The models also consider bridge weekends, in the sense that when there is a holiday on Tuesday (resp. Thursday), the Monday (resp. Friday) is also set as a holiday. We replicated the GBM modelling and tuning strategy previously used in Guevara et al., (2021) with random search in the hyper-parameter space and rolling-origin cross-validation (appropriate for time series).


The $T\_pop(d))$ is defined as follows (Eq. 2):

$$T\_pop(d) = \sum_{x=1}^{n} \frac{T_{2m}(x,d)*Pop(x)}{\sum_{x=1}^{n} Pop(x)} \qquad (2)$$

Where $T_{2m}(x, d)$ is the daily mean 2-meter outdoor temperature for grid cell x and day d [°C]; *Pop(x)* is the amount of population included in grid cell x [nº of inhabitants] and n is the total number of grid cells that corresponds to a specific country. Outdoor temperature information was obtained from the ERA5 reanalysis dataset for the years 2015 to 2020 (C3S, 2017), while gridded population was derived from the Gridded Population of the World, Version 4 (GPWv4; CIESIN, 2016).


The difference between the daily BAU and measured 2020 electricity demand levels were used to derive country-dependent daily emission adjustment factors, as described in Eq. 2:





$$EAF_{pub\_pow}(d,c) = \left(\frac{ED_{measured}(d,c) - ED_{BAU}(d,c)}{ED_{BAU}(d,c)}\right) * 100 \qquad (2)$$


where $EAF_{pub\_pow}(d,c)$ is the final emission adjustment factor for the energy industry sector for day d and country c [%]; $ED_{BAU}(d,c)$ is the estimated BAU electricity demand for day d and country c [MW] and $ED_{measured}(d,c)$ is the measured electricity demand for day d and country c [MW].

Figure 1 shows the daily adjustment factors obtained for the GNFR_A sector and selected countries (i.e., Spain, France, Germany, UK and Sweden). The resulting trends are consistent with the national lockdown calendars and levels of restriction implemented in each country. During the strictest period of the first lockdown, Italy experienced the largest reductions (-30%), followed by Spain (-25%) and France (-20%). For Sweden, positive values are observed during the same period, in line with the results reported by Le

Quéré et al. (2020). It is likely that in this country electricity demand from commercial services remained unperturbed as no national lockdowns were enforced. We also hypothesize that a voluntary self-isolation of a fraction of the population may have increased household electricity consumption. When confinement was eased, electricity demand shows the first signs of recovering in all countries. This trend is confirmed in summer, as governments softened even more lockdown measures. Italy is where the recovery is more

pronounced, reaching emissions above the BAU during August. A second significant drop of emissions is observed in France and UK and, to a lesser extent, in Italy during November 2020 coinciding with the implementation of a second round of lockdowns. Emissions rebound sharply after that, and are back to BAU levels or even above during Christmas holidays.

### 2.1.2 Manufacturing industry

The adjustment factors for manufacturing industry (GNFR_B) are based on the monthly Industrial Production Index (IPI) values reported by Eurostat (2021a). We considered the seasonally and calendar adjusted data. Note that for UK the IPI values for November and December 2020 were derived from ONS (2021) as Eurostat only reports information until October 2020 for this country. The original IPI values reported for each individual economic activity (NACE Rev. 2) were grouped and averaged into the three



subcategories listed in Table 2, according to the impacts of the COVID-19 restrictions observed on their activity:

- GNFR_B1: Manufacture of petroleum refining products. This industrial branch was considered to be essential and therefore was less affected than other industries during the full lockdown phase.

However, and due to the large decrease on the demand for finished petroleum products (e.g., jet fuel, motor gasoline), the recovery of its activity has been lower than in other sectors during the lockdown exit process.

- GNFR_B2: Manufacture of pharmaceutical, chemistry, food and beverages products. These industrial branches were also considered to be essential during the full lockdown phase, but in

contrast to the petroleum industry, the demand associated to their products barely decreased or even increased during or after the lockdown, which is translated in a low decrease (slight increase) of their activity.

- GNFR_B3: Manufacture of other products (i.e., non-metallic mineral products, basic metals, paper and paper products and machinery and equipment). These industries were considered non-

essential and therefore were heavily affected during the lockdown period as in the majority of cases were forced to close. Nevertheless, a sharp recovery is observed with the easing of lockdowns.

For illustration, Fig. 2 shows the behaviour of the IPI monthly values from January 2019 until December

2020 for each of the three aforementioned manufacturing industrial subgroups for Germany, Spain, Italy and the UK. The dashed black line represents the general IPI for the overall manufacturing industry. For the manufacturing industrial subcategories GNFR_B2 and GNFR_B3, the averaging was done considering the share of each industrial branch (i.e., pharmaceutical, chemistry, food and beverages products for GNFR_B2 and non-metallic mineral products, basic metals, paper and paper products and

machinery and equipment for GNFR_B3) to the total fossil energy final consumption as reported by the Eurostat (2021b) energy balances. For GNFR_B3, the manufacture of basic metals and non-metallic mineral products are the largest energy intensive activities (almost 70% of total energy consumption),

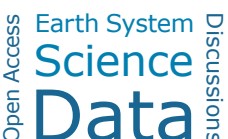

whereas manufacturing of paper and machinery and equipment represent approximately 30% of total energy consumption (Fig. S2). Note that other industrial branches originally included in GNFR_B3 (i.e.,
manufacture of wood, textiles and leather) were not considered in the final calculations since the Eurostat IPI statistics for these industrial categories are incomplete. It is expected that the removal of these industrial branches won't have a major impact on final results as their total fossil fuel consumption is not predominant (i.e., 12% in total according to Fig. S2).

For each manufacturing industry subgroup, we computed monthly and country-specific adjustment factors from a baseline taken as the average value over the two months prior to the lockdown (January and February 2020). The computed monthly adjustment factors were translated into daily adjustment factors by considering the Oxford COVID-19 Government Response Tracker dataset (OxCGRT; Hale et al., 2021). The OxCGRT provides a systematic cross-national, cross-temporal measure to understand how
government responses have evolved over the full period of the COVID-19 spread. We considered the indicator "workplace closing", which records the closings of workplaces according to four different scales of intensity: 0 – no measures, 1 – recommended closing, 2 - require closing (or work from home) for some sectors or categories of workers and 3 - require closing (or work from home) all-but-essential workplaces. We assumed that changes on industrial emissions during March started to happen in each
country once the corresponding indicator reached a value of 2 or more.

Daily emission adjustment factors were computed as a weighted average of the adjustment factors obtained for each industrial subcategory (Eq. (1)), taking into account their relative contribution to total GNFR_B emissions (Fig. 3).
Figure 1 illustrates the resulting adjustment factors proposed for NOx and NMVOC emissions, respectively. A common pattern is observed for the two pollutants, with the largest reductions occurring during April, when the restrictions were at their maximum and a large number of facilities were not allowed to operate. A pronounced recovery is observed from May onwards, coinciding with the easing of
the lockdowns and the recovery of the industrial activity. For NOx, the computed reductions are larger



than for NMVOC, with Italy, France and Spain presenting the largest decrease (between -35% and -40% during April). Low reductions are observed for Sweden, where emissions never decreased more than - 20%. Emission reductions reached levels close to BAU by the end of the year in almost all countries, as the new curfews adopted around October/November/December did not affect the manufacturing industry.

In the case of NMVOC, a general lower reduction than for NOx emissions is observed, with most countries presenting a maximum decrease below -30% during April. It is worth noting that some countries even experienced an increase in emissions during the beginning of the first lockdowns (up to 10%). The adjustments computed for NMVOC are different relative to NOx as its emissions are related to food, beverage, pharmaceutical and chemical industry branches (Fig. 3), which were less affected by the

COVID-19 restrictions or even had to increase their productivity due to an increase in demand. The largest emissions reductions are reported for Italy, and the lowest ones for UK and Sweden, with the latter even showing emission values above BAU levels (i.e., up to 5%) during the second semester of 2020.

### 2.1.3 Other stationary combustion activities

This sector includes emissions from stationary combustion activities related to the commercial and
institutional sector, the residential sector and other stationary sectors such as agriculture, forestry, fishing and military sectors.

Our emission adjustment assume that the COVID-19 restrictions only affected the combustion activities in the commercial/institutional and residential sectors. In the first case, significant emission reductions
are expected as a result of the closure of schools, universities, public buildings, restaurants, and other non-essential businesses. In the second case, emission increases are expected due to the required household confinement during the lockdown period. Regarding the agriculture, forestry and fishing sectors, we assumed no changes occurred as they were considered to be essential.

The emission adjustment factors considered for this sector are based on Google COVID-19 Community Mobility Reports (Google LLC, 2021). The Google dataset reports daily movement trends over time by geography (country and region) across different categories of places (i.e. groceries and pharmacies, parks,



transit stations, retail and recreation, residential and workplaces) based on aggregated and anonymized sets of data from users who have turned on the Location History setting for their Google Account on their

mobile devices. Reductions for each day are calculated by Google from a baseline taken as the median value, for the corresponding day of the week, over a 5-week period prior to the lockdowns (3 January to 6 February). For this sector, we used the mobility trends reported for the following categories:

•       Retail and recreation: Mobility trends for places like restaurants, cafes, shopping centres, theme

340            parks, museums, libraries, and movie theatres

•       Grocery and pharmacy: Mobility trends for places like grocery markets, food warehouses, farmers markets, specialty food shops, drug stores, and pharmacies.

•       Workplaces: Mobility trends for places of work.

•       Residential: Mobility trends for places of residence


The mobility trends for retail and recreation, grocery and pharmacy and workplaces were used to derive an average trend for the commercial/institutional sector, while the mobility trends for places of residence were used for the residential sector.

These Google trends report changes in movements, which does not necessarily represent changes in energy consumption (i.e., fossil fuels and biomass) and associated emissions. The increases in residential activity reported by Google are significantly larger than the ones reported in Le Quére et al. (2020), which indicates an average increase of 5%, and a maximum increase of 10% during the most restrictive lockdown phase. The results reported in Le Quére et al. (2020) inferred from UK smart meter data are

consistent with the ones reported by the thermostat maker Tado (Tado, 2020), which indicates an average increase of 14% in home heating consumption in Europe during March 2020 compared to March 2019. Considering the aforementioned results, the original Google trend values for the residential sector were scaled down for countries to have a maximum daily relative change of 10%. Our approach is limited by data availability and further constraints will require more data on residential energy consumption.




In the case of the commercial/institutional sector, we also adjusted the original daily decrease trends reported by Google making use of energy consumption statistics. We used information provided by IDAE (2018) on the energy consumption in the Spanish commercial/institutional sector. As shown in Table S2, Spanish commercial buildings represent more than 40% of the total energy consumption (fossil fuels and

biomass) in the commercial/institutional sector, followed by workplaces (26.5%), hospitals (11.6%), other buildings (8.8%, e.g., museums, public buildings and religious buildings), schools and universities (7.8%) and restaurants and hotels (4.3%). We hypothesized that the Spanish national lockdown restrictions implied a change in the energy consumption of: (i) -100% in schools and universities (all buildings were closed), (ii) -90% in hotels and restaurants (certain hotels were converted into temporary medical

facilities), (iii) -80% in workplaces, commercial buildings and other buildings (supermarkets and other grocery stores remained opened during the entire lockdown, as well as certain workplaces that were considered to be essential) and (iv) +50% in hospitals (due to the increase in the number of patients to attend). We combined the aforementioned information and derived an overall maximum reduction in energy consumption across Spanish commercial/institutional buildings of -66.9%. Following the

approach applied for the residential sector, we scaled up the original Google trend values for the commercial/institutional sector to set this minimum value.

Daily emission adjustment factors for the other stationary combustion sector were computed as a weighted average of the adjustment factors obtained for each GNFR_C subcategory (Eq. (1)), taking into account

their relative contribution to total emissions (Fig. 3).

Figure 1 illustrates the resulting adjustment factors proposed for NOx and PM10 emissions, respectively. For NOx, major reductions are observed for the United Kingdom, France and Italy. In these three countries, maximum reductions between -15% and -20% were reached during the strictest lockdown

period. On the contrary, and despite being under similar lockdown measures, in Spain the maximum relative reduction during the same period was only -10%. This is explained by the different contributions of agriculture, forestry and fishing subcategories (GNFR_C3) to the total GNFR_C NOx emissions. While in Spain this category represents around 40% of total NOx emissions, in France, Italy and the



United Kingdom the contribution is lower than 10% (Fig. 3). Assuming that this category was not affected

by the COVID-19 restrictions implies a lower overall emission reduction in Spain. In the case of Sweden, a slight emission increase is observed during the whole period of study. We hypothesize that this is a consequence of the likely small perturbation of the public and commercial service activity (i.e., non-essential businesses were not forced to close) and a slight increase of the residential activity as a consequence of a voluntary self-isolation of a fraction of the population. By the end of August most

countries reached or were about to reach their BAU levels, except for the United Kingdom, where emissions were still -10% below pre-lockdown values. A second significant drop in emissions is observed in France, United Kingdom and Italy during November, which is related to the forced closure of non-essential business under the second epidemic wave.

For PM10, an increase in the business-as-usual levels is observed for all selected countries. This is explained by the fact that a majority of total emissions are driven by changes in the residential sector (Fig. 3), which increased its activity due to the enforced confinement. Germany is the country that registered the lowest increase in total emissions (maximum increase of approximately 2.5%) compared to the other countries. This is again explained by the different contributions of subcategories to total GNFR_C

emissions. In this particular case, the German commercial/service subcategory represents around 10% of total emissions, while in the other countries the contribution for this subcategory is less than 5% (Fig. 3). By the end of August, all countries were close to reach the BAU levels again, and in some countries like Italy emissions levels even reached values below BAU, as people started to spend more time outdoors. A slight increase in emissions is observed during November, coincident with the introduction of new

additional mobility restrictions to curb the high incidence during the second wave of COVID-19 spread.

### 2.1.4  Fugitive emissions

This sector covers the release of emissions during the extraction and processing of fossil fuels along with their delivery to the point of final use. The activities selected for the development of specific COVID-19 related emission adjustment factors were as follows: 1) Coal mining and handling, 2) Refining / storage





& venting and flaring and 3) Distribution of oil products (gasoline). Other subcategories included in this sector were assumed to be unaffected by lockdowns and mobility restrictions.

The following sources of information were used to derive the adjustment factors:

- GNFR_D1; coal mining and handling: monthly indigenous production of hard and brown coal per

country reported by Eurostat (2021c). We computed monthly and country specific adjustment factors from a baseline taken as the average value over the two months prior to the lockdown (January and February 2020). We then averaged the resulting monthly factors per month and country and derived daily adjustment factors using the "workplace closing" reported by OxCGRT, as detailed in Sect. 2.1.2.

- GNFR_D2; refining / storage & venting and flaring: monthly IPI related to the manufacture of petroleum refining products (Eurostat, 2021a). For this subcategory, we used the same adjustment factors as for GNFR_B1 of the manufacturing industry (see Sect. 2.1.2).

- GNFR_D3; distribution of oil products (gasoline): we assumed that changes in this activity can be represented by changes in road fuel sales in filling stations, which at the same time can be

linked to changes in road traffic activity. This hypothesis is illustrated in Fig. S3, which shows the relationship between monthly/weekly changes in petrol sales and traffic activity for selected countries. In all cases the Pearson correlation coefficient (PCC) is larger than 0.9, the intensity in the drop of petrol sales during the lockdown periods fairly coinciding with the decrease in traffic activity. Considering these results, for this activity we used the same emission adjustment factors

for road transport gasoline exhaust emissions (see Sect. 2.1.6).

GNFR sector-level daily emission adjustment factors were computed as a weighted average of the adjustment factors obtained for each subcategory (Eq. (1)), taking into account their relative contribution to total GNFR_D emissions (Fig. 3).


Figure 1 shows the adjustment factors for NMVOC fugitive emissions from fossil fuels. The pattern of emission decreases is significantly different from one country to another, mainly because of the effect of



the individual subcategory that dominates total emissions in each country, and to a lesser extent due to the different levels and types of restrictions implemented. For instance, in UK almost 40% of total

NMVOC emissions come from refining activities (storage, flaring) and therefore the decrease in emissions is largely driven by their decrease (Fig. 3). On the other hand, approximately 50% of total NMVOC emissions in France comes from the distribution of oil products, and subsequently the drop in emissions is similar to that of road traffic emissions, with two significant drops corresponding to the lockdowns implemented during the Spring and Fall epidemic waves.

**2.1.5   Use of solvents**

The GNFR_E category includes NMVOC emissions coming from the residential/commercial and industrial use of solvents. Our assumption for this sector is that the COVID-19 restrictions only affected certain industrial subcategories, including: (i) GNFR_E1: the use of organic solvents to remove grease, fats, oils, wax or soil from metal products and (ii) GNFR_E2: the use of inks in the printing industry.

Other industrial activities that involve the use of solvents (e.g., manufacturing of pharmaceutical products or automobiles) could not be considered as they are not individually distinguished in the NFR reporting nomenclature, but rather reported as part of broader categories (e.g., 2.D.3.g: Chemical products, 2.D.3.i: Other solvent use, 2.G: Other product use). Emissions from domestic and commercial solvent use were assumed to remain constant due to the lack of specific activity data to compute the adjustment factors and

the limited number of categories considered in the NFR nomenclature. We hypothesize that the potential increase in the use of certain products containing solvents, such as cleaning products, was compensated by the potential decrease in the use of other products, such as car products or cosmetics for personal care. We are aware that this hypothesis may be limited by the increased use of the so-called "pandemic products" triggered by the COVID-19 (Steinemann et al., 2021), which includes products intended to

clean and disinfect, such as hand sanitizers or surface cleaners. However, the lack of specific information does not allow us computing associated adjustment factors.

The adjustment factors for industrial solvent use are based on the monthly IPI values adjusted for seasonal and calendar effects (Eurostat, 2021a). As already mentioned in Sect. 2.1.2, for UK the IPI values for

November and December 2020 were derived from ONS (2021). The "Manufacture of fabricated metal products, except machinery and equipment" and "Manufacture of computer, electronic and optical products" on the one hand, and the "Printing and reproduction of recorded media" on the other hand were the industrial branches considered to quantify the impacts of restrictions on each of the two subcategories considered. For each subcategory, we computed monthly and country specific adjustment factors from a

baseline taken as the average value over the two months prior to the lockdown (January and February 2020). The computed monthly adjustment factors were translated into daily adjustment factors by considering the "workplace closing" reported by the OxCGRT, as detailed in Sect. 2.1.2.

Daily emission adjustment factors for the use of solvents sector were computed as a weighted average of

the adjustment factors obtained for each subcategory (Eq. (1)), taking into account their relative contribution to total GNFR_E emissions (Fig. 3).

Figure 1 illustrates the resulting adjustment factors proposed for NMVOC emissions. Decrease in emissions is generally low (i.e., below -10%) and mainly occurring during the Spring lockdowns. The

small reductions are due to the limited contribution of metal cleaning and printing industrial activities to the overall emissions from this sector (Fig. 3). A pronounced recovery is observed from May onwards, coinciding with the easing of the lockdowns and the recovery of the industrial activity.

### 2.1.6   Road transport

The emission adjustment factors considered for this sector are based on the Google COVID-19

Community Mobility Reports (Google LLC, 2021). We used the mobility trends reported for the transit stations category, which includes places like public transport hubs such as subway, bus, and train stations. We compared the Google movement trends against trends derived from measured traffic counts reported by 18 European national road administrations. Table A1 summarises the countries covered, sources of information and characteristics of the traffic count datasets considered, as well as the baseline considered

to derive traffic activity trends.



Figure 4 shows the results of the intercomparison at the country level for selected countries. Black lines represent the Google mobility trends, while read and blue lines represent the measured-based trends computed for Light duty vehicles (LDV) and heavy-duty vehicles (HDV). Similar patterns are observed 500  for all cases as a function of the period of study:

- First COVID-19 lockdown period (mid-March until mid-May): The Google dataset is capable of reproducing the LDV measured-based trends. Overall, the average reductions reported by each of the two datasets are fairly similar, with Google reporting in some cases reductions slightly larger 505  than the measured ones, particularly in Scandinavian countries (e.g., Finland, Sweden, Norway). On the other hand, a large discrepancy is observed between Google results and the HDV measured-based trends, the former presenting larger reductions. In the UK for instance, the average reduction for HDV was of -35.6% between March and 26 April, almost two times lower than the one reported by Google (-69%, respectively).

- COVID-19 lockdown-exit process (mid-May until end of September): Differences between LDV and Google trends become larger, showing different rates of recovery. Google tends to underestimate the observed recovery of traffic activity. The discrepancies between measured trends and the Google dataset become larger with time. During summer (i.e., July, August), the LDV trends in the majority of countries are close or even above business-as-usual levels (e.g., 515  Netherlands, Ireland), yet Google continues to report mobility values that are below business-as-usual levels. In the case of HDV trends, discrepancies with Google trends are reduced but still significant.

- Second COVID-19 lockdown period (beginning of October until end of December): Discrepancies between Google trends and LDV/HDV measured-based trends remain almost 520  unchanged. Google trends are, qualitatively speaking, capable of reproducing the drops in traffic activity observed in the LDV measured-based trends during November and the Christmas season, but not quantitively speaking, as reductions are systematically larger than the observed ones.



Figure 5 shows a comparison between averaged monthly adjustment factors for road traffic reported by
Google LLC (2021) and LDV measured-based trends per each of the countries listed in Table A1.
Discrepancies between Google and measured-based trends started to increase with the easing of the
restrictions (May) and reached a maximum difference in September. During this month, the average
traffic reductions reported by the LDV measured-based trends are in a range between -20% and 0%, while
in the case of Google reductions are between -40% and -10%. The overestimation of Google reductions
when compared to measured-trends is somewhat reduced during November and December, coinciding
with the implementation of new lockdowns, but large differences are still observed. The plot also shows
how the decrease of traffic activity during April (first lockdown) was much larger than during the second
wave of restrictions (i.e., November): In April, maximum traffic reduction was - 80% (Spain, ES), while
in November the maximum drop was around -35% (France, FR).

The differences observed between measured-based trends and the Google trends are mainly related to the
fact that Google data refers to mobility trends in public transport hubs. As a result of COVID-19, people
are now avoiding public transport as these can be considered places where it might be difficult to avoid
contact with other passengers (De Vos, 2020). The adjustment factors proposed by Google during the
lockdown exit process are affected by this factor and therefore are underestimating the observed changes
in traffic activity during the lockdown exit process. This hypothesis is illustrated in Fig. S4, where the
traffic movement trends obtained in Rome are compared to the evolution of the access to subway stations.
The recovery of mobility in the subway system during the lockdown exit process is very much in line
with the Google trend and much lower than the one observed for the private transport sector. On the other
hand, the lower reduction observed in HDV's activity when compared to Google is because these vehicles
supported the delivery of essential goods and products during the confinement (e.g., food, medical
supplies), and subsequently their use decreased much less than that of LDV.

In order to overcome the identified limitations of the original Google trends, we used the LDV and HDV
measured-based trends compiled for the different countries to produce two sets of European correction
factors: (i) HDV correction factors and (ii) LDV correction factors. In both cases, the correction factors





were computed as the ratio between the weekly average changes in traffic activity reported by the measured trends and the weekly average changes in mobility reported by Google. The resulting country-level weekly correction factors were then averaged to obtain a set of European weekly correction factors.

The countries considered to develop the European average weekly correction factors were the ones listed in Table A1 except Poland and Estonia, as the number of traffic stations used to derive measured-based trends for these two countries was small.

The two sets of correction factors were applied to the original Google mobility trends in order to derive

two new sets of adjustment factors for LDV and HDV emissions. Note that for those countries for which we had daily traffic count datasets available (i.e., United Kingdom, Norway, France, Spain, Finland, Ireland, Netherlands and Switzerland) we directly substitute the original Google trends for the ones derived from traffic counts. Similarly, for countries with weekly and monthly traffic count datasets, adjustments of the original Google trends were done by considering only the correction factors of the

corresponding country.

We applied the adjusted Google transit mobility trends with the LDV factors to the GNFR_F1 (exhaust gasoline) and GNFR_F3 (exhaust LPG gas) sectors, as the contribution of HDV to their emissions is null or almost residual. However, for the GNFR_F2 (exhaust diesel) and GNFR_F4 (non-exhaust) sectors, the

final emission adjustment factors were computed as a weighted average of the adjustment factors obtained for LDV (GNFR_F21 and GNFR_F41) and HDV (GNFR_F22 and GNFR_F42) vehicle categories following Eq. (1) and considering their relative contribution to total corresponding emissions (Fig. 3).

Figure 1 illustrates the adjustment factors for road transport diesel exhaust (GNFR_F2) NOx and $CO_2$_ff

emissions. The patterns of the emission adjustment factors for the two species are very close. However, the reductions reported during the Spring lockdowns (March and April 2020) are slightly lower for $CO_2$_ff, especially in countries where the HDV emissions have a larger contribution to total emissions such as Spain, Italy and France (Fig. 3). The decrease of the traffic activity in Italy started two days after the implementation of the localized lockdown (23 February) and intensified once the national lockdown





was imposed on 12 March, reaching reductions of about -80%. In the case of Spain and France, similar traffic reduction levels were reached just 3 days after the beginning of the corresponding national lockdowns. For the UK and Germany, the largest reductions are around -70% and -50%, respectively. The smaller reductions in Sweden (around -40%) are consistent with the lack of enforced mobility restrictions in this country at any point. In all cases, the activity started recovering during the last week

of April, coinciding with the relaxation of the mobility restrictions. This trend is confirmed between May and August, with a steady recovery observed in all countries except for Spain, where a slight decrease occurs during July. This abrupt change in the upward trend corresponds to a sudden increase of infections in this country and the subsequent implementation of additional measures to restrict mobility. In contrast, large recovery rates were observed in Italy, Germany and UK, where values even exceeded BAU levels

during certain days in July and August. However, the introduction of new restrictions measures continued to curb traffic activity in October. Strengthening measures caused a second significant drop in emission during November, although it was ~50% lower than that of April (e.g., UK, Italy). The first weeks of December were marked by a relaxation of the second lockdown measures and a subsequent recovery of the traffic emissions. However, a third drop in emissions was observed during the Christmas season, as

additional measures were implemented to restrict social gatherings.

### 2.1.7   Aviation

We derived the adjustment factors related to air traffic emissions during Landing and Take-Off cycles (LTO) in airports from statistics provided by EUROCONTROL (2021), which reports daily arrivals and departures by airport from January 2016 to December 2020. We computed day and country-specific flight

operation reductions from a baseline taken as the average value for the corresponding day of the week (Monday to Sunday and National holidays) and month of the year from 2019.

Figure 1 illustrates the resulting emission adjustment factors for selected countries. For most countries the reductions in flight activity started some days before the implementation of the national lockdowns,

as certain international flights (especially the ones coming from and going to Asia) were already being cancelled. It is observed that in almost all countries, the reduction levels reached values of -90% or more



before the beginning of April. In contrast to road transport, the signs of recovery during May and June are very weak as the movements between countries were still restricted at that time. On the contrary, a general more pronounced recovery was observed during July and August as a consequence of the 610 beginning of summer holidays and the lifting of restrictions to travel. This recovery was especially significant in Spain and France. However, most of the countries still presented reductions larger than -50% during summer. Strengthening measures linked to the second wave of infections negatively impacted European air traffic in November, when new drops were observed, especially in the UK and France. The end of the year, however, was marked by a recovery in air traffic operations, similarly to the one observed 615 during summer, that can be attributed to the Christmas holidays.

### 2.1.8 Shipping

Emission adjustment factors for the shipping sector were based on the AIS-based gridded emissions computed by the STEAM model (Jalkanen et al., 2012 and 2016) under CAMS (Granier et al., 2019). Weekly and sea-region dependent adjustment factors were derived as the ratio between the shipping 620 emissions reported for a given week in 2020 and the emissions reported by the equivalent week in 2019. Estimated $CO_2$ emissions were used as a proxy to compute the adjustment factors, as this pollutant can give a more direct indication of the changes in the fuel used. The use of other pollutants such as $SO_2$ or PM would mask the impact of COVID-19 on 2020 emissions, as they were affected by the implementation of global 0.5% sulphur cap, as discussed in Sect. 3.


Figure 1 illustrates the adjustment factors produced for selected sea regions (i.e., Atlantic Ocean, ATL; Baltic Sea, BAS; English Channel, ENC; Mediterranean Sea, MED; North Sea, NOS and Norwegian Sea, NWS). In general terms, the decrease in shipping emissions began in week 12 (i.e., 16-22 March) and followed a downward trend until mid-June. From that point, a slight constant recovery was observed in 630 most sea regions, with sporadic ups and downs (e.g., NWS). By the end of the year, some sea regions were already close to BAU levels (e.g., BAS, -5%). Overall, MED and NOS were the sea regions presenting the largest (i.e., -17%) and lowest reductions (i.e., -3%), respectively. The contrast in the results obtained for these two sea regions is very much related to the different contribution of passenger



ships to total shipping traffic, which is larger in the MED than in NOS. As reported by EMSA (2021),
cruise ships and Ro-Ro/passenger ships were the ship types mostly affected by COVID-19, showing
reductions of 2020 ship calls in EU ports of -85% and -39% when compared to 2019 levels. These
reductions were significantly larger than the ones reported for cargo ships (between -7% and -2%), which
are dominant in NOS.

### 2.1.9 Off-road transport

The GNFR_I category reports emissions from non-road mobile machinery that is used in several sectors,
including: (i) commercial (e.g., transportable equipment), (ii) residential (e.g., gardening and handheld
equipment), (iii) agriculture/forestry/fishing (e.g., harvesters, cultivators), (iv) manufacturing industries
and construction (e.g., excavators, loaders, bulldozers) and (v) other categories including military, land-
based railways and recreational boats. In the present work, the impact of COVID-19 restrictions was
quantified for emissions related to mobile machines in the manufacturing industry and construction sector
(GNFR_I1), while emission from the other subcategories (GNFR_I2) were assumed to remain unaffected.

The adjustment factors are based on seasonally and calendar adjusted monthly IPI values reported by
Eurostat (2021a). We considered the IPI values reported for the general manufacturing and construction
categories. As for the manufacturing industry, monthly and country specific adjustment factors were
computed taking as a baseline the average value over January and February 2020. The translation from
monthly to daily factors was done considering the evolution of the "workplace closing" indicator reported
by OxCGRT.

Figure 1 shows the emission adjustment factors for NOx emissions. The decrease in emissions is generally
low, with a maximum reduction of less than -15% in UK during April, and reductions between -2.5% and
-5% in Germany and Spain during the same period. As showed in Fig. 3, the contribution of the
manufacturing industry and construction machinery subcategory to total emissions is rather low (30% in
average at the EU27 + UK level), which explains why reductions are not as large as the ones showed in
e.g., the GNFR_B manufacturing industry sector. Emissions are reaching levels close to BAU by the end





of the year in almost all countries, as the new virus-related curfews adopted during the second wave did not affect the industrial manufacturing and construction activities.

## 3    Business-as-usual 2020 emissions

Our adjustment factors were designed to be applied to a gridded emission BAU inventory for 2020

developed based on the CAMS European regional emission inventory (CAMS-REG_v5.1) time series, ranging from 2000 to 2018 (update from Kuenen et al., 2021). The CAMS-REG_v5.1 dataset makes use of official air pollutants and greenhouse emission inventories submitted by each country to EMEP, UNFCCC and the EU. Those country-level annual data form the basis of the emission inventory and are spatially disaggregated to a $0.1° \times 0.05°$ grid for use in chemical transport models. Besides the 12 GNFR

sectors for which the COVID-19 adjustment factors are prepared (Table 1), the inventory also includes emissions from waste management (GNFR_J), livestock (GNFR_K) and other agricultural activities (GNFR_L). Additional sub-sectors are also defined, as explained before in Sect. 2. The methodology applied and sources of information used for the construction of the CAMS-REG emission inventory are described in detail in Kuenen et al. (2021)


The main disadvantage of the CAMS-REG_v5.1 gridded inventory is the 2-year lag in emission reporting. To overcome this limitation a method was developed to estimate emissions for recent years (y-1), which makes use of sector-specific activity data. We have updated this methodology to make a BAU emission estimate for 2020 to be combined with the COVID-19 adjustment factors described in Sect. 2. The method

follows three steps:

- Estimate the activity data (AD) per sector, country and year. For this we gathered data from a range of sources, which are listed in Table 3. If activity data are available for 2020 we use it directly. Otherwise, if activity data are available for previous years (time series cover between 7

685        to 21 years for the different data sources) we examine whether a significant trend exists ($R^2 > 0.3$) and extrapolate that to 2020.



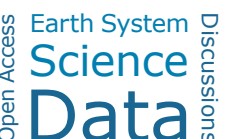

- Estimate the emission factor (EF) per sector, country, year and pollutant. The emission factor is calculated by dividing the emissions for 2000-2018 by the AD. Again, if a significant trend in EF exists ($R^2 > 0.3$) we extrapolate that to 2020. Otherwise, the EF of the last reporting year is used (here: 2018).

- Finally, we calculate the emissions for 2020 by multiplying AD and EF. If AD is missing this gives no result. In that case we examine whether a significant trend exists ($R^2 > 0.3$) in the emission time series of 2000-2018. If so, it is extrapolated to get an emission estimate for 2020. Otherwise, the emission of the last reporting year is used (here: 2018).

Additionally, we have included the impact of the 0.5% sulphur cap on (international) shipping fuels as of January 1, 2020 (IMO, 2019). For the North Sea, Baltic Sea and English Channel we assume no impact of the sulphur cap, as these sea regions are part of the Sulphur emission control areas (SECA) and already showed strong reductions before (Kattner et al., 2015). For all other sea regions, we assume a 75% reduction in $SO_2$ emissions compared to 2018. Also for PM we assume a 48% reduction compared to 2018 due to the reduction of $SO_4$.

For the 2020 BAU emission estimates we ignore all AD that is impacted by the COVID-19 lockdowns and mobility restrictions. We still use the AD for trend analyses though, as a trend caused by, for example, technological progress will continue in 2020 and therefore be part of the BAU emission estimates. Note that not all GNFR sectors are included in Table 3, for example due to absence of AD. In that case the emissions from 2018 are copied to 2020.

Figure 6 shows the $NO_x$ emission time series for Italy and Sweden from 2010 to 2020, where 2020 represents the BAU estimate. The percentages indicate the difference compared to 2018, which are caused by normal trends in activity and emission factors. We also provide an estimate where we do include AD affected by COVID-19 (separate bar). We find that $NO_x$ emissions from road transport decreased since the start of the time series, but COVID-19 causes an even stronger decrease in emissions compared to 2018. The same is true for public power and manufacturing industry, although the trend in Sweden is

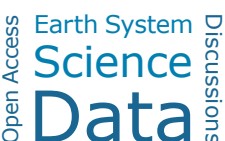

weaker and also the COVID-19 impact on the manufacturing industry is less. Emissions from other stationary combustion activities show an increase in 2020 in Italy (+5%), because it was a bit colder than in 2018. In Sweden, 2020 was warm compared to 2018 and the opposite effect is visible (-15%). This estimate is not affected by COVID-19, because it is purely based on the temperature (i.e., changes in the yearly degree days). Note that the estimate with COVID-19 is not comparable to the adjustment factors,

as the AD used here do not necessarily capture the impact of the lockdowns. We merely use it to illustrate that the BAU estimate indeed represents a situation without COVID-19.

## 4    Discussion of the emission changes

This section presents the estimates of daily sector-, pollutant- and country-specific European emissions from January 1$^{st}$ to December 31$^{st}$ 2020 and compares them to the levels of emissions as expected in the

BAU scenario described in Sect. 3. Emissions for 2020 (hereafter referred to as COVID-19 scenario) were obtained as a combination of the original CAMS-REG-v5.1 2020 BAU annual gridded emissions and the emission adjustment factors presented in Sect. 2. The original CAMS-REG-APv5.1 2020 BAU annual emissions were broken down into daily resolution using the sectoral dependent emission temporal profiles reported by Denier van der Gon et al. (2011). For the COVID-19 emission scenario, the emission

adjustment factors were combined with these temporal profiles in order to model dynamic emission changes for each sector and country, as described in Guevara et al. (2021). The analysis of the results focuses on multiple aspects of the COVID-19 restrictions on emissions, including a description of the temporal evolution of emissions at the EU27 + UK level, per country, species and pollutant sector, as well as an analysis of the spatial distribution of the changes in total emissions.

### 735    4.1.1    European and country-level analysis

Figure 7 illustrates the COVID-related changes in the EU27 + UK daily emissions for criteria pollutants and greenhouse gases (GHGs) between January 1$^{st}$ and December 31$^{st}$ 2020 as compared to the BAU scenario. Dotted and solid lines represent the BAU and COVID-19 daily emissions, respectively, and differences between them are illustrated with the shaded areas.




For all pollutants, the decrease in emissions started to occur during the first weeks of March, coinciding with the implementation of localized and national lockdowns to reduce mobility and social interactions. The greatest reductions are observed at the end of March and beginning of April, when the restrictions were at their maximum. During late April and the beginning of May, emissions began to recover in a

persistent and continuous way, as national governments started to roll-back COVID-19 measures and the different economic activities resumed. By mid-September emissions of all pollutants were close to reaching again pre-lockdown levels. However, a second drop in emissions similar to that of June is observed during end of October and beginning of November, coinciding with the implementation of a new round of mobility restrictions to break the second wave of COVID-19 infections. Reductions in

emissions remained almost unchanged until the end of the year, as restrictions were kept in place with a few exceptions during the Christmas holidays. It is important to note that the daily evolution of the emissions plotted in the charts is not only affected by the COVID-19 restrictions, but also by the inherent seasonality associated to emissions from each pollutant sector. For instance, emissions from other stationary combustion activities are mainly related to the combustion of fuels in households/commercial

buildings for space heating, and therefore they decrease as winter ends and outdoor temperatures start to be higher. This fact can be observed with the daily evolution of $PM_{2.5}$ and $CO_2\_bf$ emissions, as they are mainly driven by residential wood combustion emissions.

In the aggregated, a reduction of -10.5% (-602 kt) was seen in $NO_x$ emissions, followed by a -7.8% (-

260.2 Mt) in $CO_2\_ff$, -4.7% (-808.5 kt) in CO, -4.6% (-80kt) in $SO_2$, -3.3% (-19.1 Mt) in $CO_2\_bf$, -3.0% (-56.3 kt) in $PM_{10}$, -2.5% (-173.3 kt) in NMVOC, -2.1% (-24.3 kt) in $PM_{2.5}$, -0.9% (-156.1 kt) in $CH_4$ and -0.2% (-8.6 kt) in $NH_3$. The largest decline in European emissions was observed during the month of April for all pollutants, with an abrupt -32.8% and -25.5% decrease in total $NO_x$ and $CO_2\_ff$ emissions, which corresponds to -157.3 kt and -70.2 Mt, respectively (Fig. 8). Around 25% of the total drop in

emissions occurred in 2020 took place during the month of April. As mentioned before, emission levels in September were already close to the pre-lockdown levels, although still presenting a slight decrease when compared to the BAU scenario (-4.8% and -3.9% for $NO_x$ and $CO_2\_ff$, respectively). The emission reductions observed during November and December (i.e., up to -10.5% for $NO_x$ and -6.5% for $CO_2\_ff$)



were lower than those that occurred during the first epidemic wave because mobility restrictions
implemented by governments were generally slower and softer (e.g., curfews, limited social gatherings,
early closing times for restaurants and bars) and only had to be toughened in those countries affected by
a new and more contagious variant of the COVID-19 such as France, Germany, the UK and the
Netherlands.

Results shown in Fig. 7 and Fig. 8 allow illustrating the heterogeneous impact of the COVID-19
restrictions on total emission changes across pollutants. Worth noting is the large contrast between
decreases in NOx (-10.5%) and $PM_{10}$/$PM_{2.5}$ (-3% and -2.1%) emissions (see Sect. 4.1.2 for further
discussions). The almost null reduction reported for $NH_3$ and $CH_4$ emissions is linked to the fact that the
large majority of these emissions are related to agricultural and waste management activities (e.g., use of
fertilizers, manure management and livestock), which in the present work were supposed to remain
unaffected during the COVID-19 restrictions. This assumption is in line with the results published by
Elleby et al. (2020), which indicate that COVID-19 implied a reduction of direct GHGs from agriculture
of only about 1% at the global scale.

Figure 9 and 10 show the relative decline (%) in total emissions per country and species for criteria
pollutants and greenhouse gases, respectively. Vertical lines indicate the average relative declines
computed at the EU27 + UK level for each species. Non-shaded marks highlight those countries/species
where reductions were larger than the ones observed at the EU27 + UK level. A large variation in the
relative declines of emissions is observed between countries due to 1) the heterogeneous levels and types
of restrictions implemented across countries, and 2) the different contributions of each pollutant sector,
particularly of the road transport sector and other stationary combustion activities, to total emissions in
each country.

The most pronounced declines occur for $NO_x$ and $CO_2$ fossil fuel emissions, Italy being the country where
these two pollutants suffered the largest relative reduction (i.e., -15.1% and -11.4%, respectively). On the
other hand, Malta presents the largest relative reductions of $SO_2$, CO, NMVOC and $CO_2$ biofuel emissions



(between -17.2% and -6.8%). Despite not being among the countries where the strictest lockdowns and containment strategies took place, the contribution of road transport to total CO, NMVOC and $CO_2$ biofuel emissions in this country is significantly larger than what it is reported at the EU27 + UK level (i.e., 54.1% versus 14.8%, 87% versus 21.1%, 40.3% versus 7.5% and 69% versus 10.5%, respectively). A similar situation is observed in Cyprus, which presents the largest relative reduction of total $PM_{2.5}$ emissions (-6.2%). This country reports the lowest fraction of $PM_{2.5}$ emissions from other stationary combustion activities (4.9% versus 52.1% at the EU27 + UK level), a sector that suffered an increase in emissions during lockdown restrictions (see Sect. 4.1.2). For $PM_{10}$ emissions, the largest relative drop occurs in the UK (-6.5%), which is among the countries that suffered the strictest restrictions. In the case of $CH_4$ the largest reduction is observed in Romania (-4.1%) mainly due to the decrease of emissions from coal mining activities. Finally, for $NH_3$ most of the EU countries present relative reductions close to the average value and that are almost negligible (between -0.56% and -0.03%), as in all of them agricultural activities, which remained unaffected by COVID-19 restrictions, represent more than 90% of total $NH_3$ emissions. Results also show that for certain countries and species, emissions not only decreased but, in some cases, slightly increased due to the COVID-19 restrictions. This is the case, for instance, of $PM_{2.5}$ emissions in Hungary and $CO_2$ biofuel emissions in Croatia (i.e., 0.4% in both cases). The observed increase in these two countries is a direct consequence of the large contribution of the other stationary combustion activities to total $PM_{2.5}$ and $CO_2$ biofuel emissions, respectively. In Hungary, this sector represents 81.3% of total $PM_{2.5}$ emissions, whereas in Croatia it represents 79.9% of total $CO_2$ biofuel emissions. These values are much larger than the average contribution observed at the EU27 + UK level, which is 52.1% and 39.1%, respectively.

### 4.1.2 Sectoral analysis

Figures 11 and 12 show the relative decline in emissions of criteria pollutants and GHG by sector and species in 2020 for EU27 + UK, while Fig. 13 illustrates the daily evolution of $NO_x$ emission differences per sector between January 1st and December 31st 2020.



The aviation sector presents the largest drop among all sectors, with a reduction of between -51 and -56% in emissions during 2020. The second most affected sector is road transport, which presents a decline in
emissions between -15.5% and -18.8%, depending on the pollutant. These two are by far the sectors affected the most by the COVID-19 restrictions, with $NO_x$ emission declines reaching approximately -90% and -60%, respectively, during April (Fig. 13). Despite showing drops of similar intensity, the recovery of emissions differs significantly between these two sectors. For road transport emissions started to gradually and steadily recover during late April and almost reached again BAU levels by September
(i.e., approximately -5% for $NO_x$). On the other hand, the drop in emissions from aviation remained almost unchanged until the beginning of July, when a modest rebound is observed coinciding with the beginning of the summer holidays. The introduction of new restrictions measures continued to curb road traffic activity in October. Strengthening measures caused a second important drop in November, although a ~50% lower than in April. Strengthening measures linked to the second wave of infections also impacted
tbe European air traffic emissions in November, when new drops are observed. The end of the year, however, was marked by a new slight recovery in emissions, similarly to the one observed during summer, and that can be attributed to the Christmas holidays.

For the manufacturing industry and other stationary combustion activity sectors, a heterogenous impact
of the COVID-19 restrictions is observed across the different pollutants. For the first sector, a lower reduction is observed for NMVOC and $NH_3$ (between -2.8% and -3.5%) when compared to the other pollutants (between -6.8% and -7.2%). This is due to NMVOC and $NH_3$ emissions being mostly driven by processes occurring in the food/beverage and chemistry industries, which were considered essential during the lockdown phase and were therefore less affected than other industry branches, such as the
manufacturing of basic metals or mineral products (see Sect. 2.1.2). Similar to road transport, the largest drop in industrial emissions was reported during April (i.e., -25% for $NO_x$), when a significant number of facilities were not allowed to operate. However, emissions began to recover in late April and May, as industrial activities fully resumed in large part of Europe. As shown in Fig. 13, emissions from this sector quickly picked up again, approaching its pre-pandemic levels of activity during November (i.e., -1.1% for $NO_x$). The reason for this rapid recovery is the fact that, unlike other sectors such as road transport
for $NO_x$). The reason for this rapid recovery is the fact that, unlike other sectors such as road transport





that were more limited by the measures to curb the second wave of infections, since spring there have been hardly any restrictions directly affecting manufacturing industrial activities.

For the other stationary combustion activities, the pollutants that are mainly related to residential wood
combustion processes (i.e., $PM_{10}$, $PM_{2.5}$, $NH_3$, NMVOC, CO, CO2_bf and $CH_4$) experienced a slight increase (between 1.1% and 1.7%), while the rest of pollutants (i.e., $NO_x$, $SO_2$ and CO2_ff) showed a modest decrease (between -0.4% and -2.9%). In both cases, the cumulative changes were not substantial and after the lockdowns in Spring, emissions were practically back to BAU levels by the end of July 2020 (Fig. 13). A new decrease in emissions is observed during November and December, coinciding with the
new round of restrictions and the closure or limitation of working hours of non-essential commercial business such as restaurants or shopping stores.

In the public energy sector, the overall relative reduction in emissions during 2020 was approximately -3.3%. As for the previous sectors, large differences are observed between months: In September, public
energy emissions in the COVID-19 scenario were only -2.5% lower than in the BAU scenario, compared to being -12% lower in April. As in the case of the manufacturing industry sector, emissions were barely affected during the Fall restrictions and were almost back to BAU levels during December.

The shipping sector experienced a decrease in emissions of around -9.5% for all pollutants. The evolution
of daily emissions in this sector indicates a slow recovery of the activity, which is partially linked to the slow recovery of maritime passenger services. Decrease in emissions from off road transport emissions were between -3% and -1.8%. More than 50% of the total drop in emissions from this sector happened between April and May, when restrictions were at their maximum. After this period, a rapid recovery is observed, emissions being only -1% below BAU by the end of the year. Fugitive emissions from fossil
fuel production and transportation show decreases of up to -10% for NMVOC and -6.7% for $CH_4$. Finally, the decrease of NMVOC emissions from use of solvents is very limited (-1.3%) as only metal cleaning and printing industrial activities were considered to be affected by COVID-19 restrictions.





The stacked area charts shown in Fig. 14 illustrate the changes in average $NO_x$ and $PM_{2.5}$ weekly emissions [t·week$^{-1}$] from individual sectors across time for EU27 + UK countries. The charts consist of multiple lines drawn to track the emission changes for various pollutant sectors, and the area below each line is coloured to represent the associated sector: Road transport (equivalent to GNFR_F), other stationary combustion activities (equivalent to GNFR_C), public energy (equivalent to GNFR_A), industry (equivalent to GNFR_B), aviation (equivalent to GNFR_H) and others (sum of emissions from all other sectors). Note that shipping emissions are not included in the results as they are not linked to any specific EU27 + UK country. A black solid line is used to represent the evolution of total emissions during the COVID-19 pandemic, and a dashed grey line is used to represent the evolution of total emissions under the BAU scenario.

The comparison between the charts produced for $NO_x$ and $PM_{2.5}$ allows understanding the heterogeneous impact of COVID-19 across pollutants presented in Sect. 4.1.1. As shown in the charts, these differences are mainly due to the fact that total emission changes were primarily driven by changes in road transport and other stationary combustion activities and the contribution of these two sectors to total emissions of each pollutant. In the case of $NO_x$, road transport is the largest contributor to total emissions and therefore the drop in total emissions is significant, while in the case of $PM_{2.5}$ the main contributor to total emissions is other stationary combustion activities, which were practically not affected by the COVID-19 restrictions. As a matter of fact, more than 70% of the total drop in $NO_x$ emissions occurred at the EU27 + UK level comes from the road transport sector.

### 4.1.3 Spatial analysis

Figure 15 shows a map of cumulative $NO_x$ emission declines [kg·cell$^{-1}$] between January 1$^{st}$ and December 31$^{st}$ as compared to the BAU scenario. The gridded emission results are provided at the same resolution as the CAMS-REG_v5.1 BAU inventory (i.e., 0.1x0.05 degrees). The main reductions occurred in urban areas and main interurban roads, especially within the most affected countries (i.e., Italy, Spain, France, and the United Kingdom). The pattern of the spatial emission difference is in line with the fact that most of the $NO_x$ emission reductions are related to road transport, as previously shown



in Fig. 14. Isolated and large emission drops can also be distinguished in certain grid cells (e.g., northwest of Spain or in the North Sea), which correspond to the decrease of emissions from individual industrial point sources.

Fig. 15 illustrate the average and 5th and 95th percentiles (p05, p95) of the daily relative changes [%] in the gridded $NO_x$ emissions for Italy and Germany, respectively. The results were computed considering all the grid cells within each of the countries. In Italy, the last 2 weeks of March and first 2 weeks of April show certain areas of the country reaching reductions up to −70 %, whereas in other areas less affected by anthropogenic (and particularly road transport) emissions the reductions were significantly lower

(∼−20%). During summer the range of relative changes becomes much lower, emissions ranging between -10% below and 10% above BAU levels. These results are in line with the fact that during this period mobility restrictions were lifted and traffic activity reached values above BAU levels due to the increase of domestic tourism. The drop in emissions observed during November and associated to the second round of nationwide COVID-19 restrictions show relative changes between ∼-26.5% and ∼-7.2%, which are

approximately two times lower than the ones observed during the first round of lockdowns. In the case of Germany, the relative changes during the lockdowns of Spring ranged approximately between −40 % (p95) and −10 % (p05). Similar to what is observed for Italy, during summer the relative decline of $NO_x$ emissions is considerably reduced, ranging between -10% and 5% below and above BAU levels, respectively. A second significant drop in emissions is observed during the second half of December,

when Germany had to go into a new hard lockdown as the number of deaths and infections from COVID-19 reached record levels. During this period of time, average emission reductions reached values of between -4.5% (p05) and up to -24.5% (p95). As in the case of Italy, the reductions associated to the second round of restrictions is approximately two times lower than the ones observed during the Spring wave.


Fig. 16 illustrates the relative $NO_x$ and NMVOC emission declines occurred in European high-density cluster urban centers, which are defined as urban regions with a density of at least 1 500 inhabitants per $km^2$ and a minimum population of 50 000. The discrimination of the CAMS-REG-AP/GHG gridded



domain between urban and rural areas was derived from the Global Human Settlement Layer (GHSL)

project (Pesaresi et al., 2019). The decline of $NO_x$ urban emissions was in average 3.4 times larger than the one obtained for NMVOC (i.e., -11.3% versus -3.3%). These results coincide with the general increase of $O_3$ levels in urban areas observed during the spring COVID-19 lockdowns, which is attributed to the fact that the $O_3$ production is largely VOC-sensitive across European urban areas (Grange et al., 2021; Querol et al., 2021). The largest differences between the $NO_x$ and NMVOC emission declines was found

in Spain (-15.6% versus -3.1%) and Portugal (-17.1% versus -3.9%). These results are in line with the relative changes in $O_3$ concentrations in traffic stations reported by Grange et al. (2021), which show that the largest $O_3$ increases occurred in Spain (61.9%) and Portugal (46.8%).

## 5    Data availability

Emission adjustment factors per country-, day of the year, sector and pollutant are provided in an Excel

file through the CAMS Document Repository (https://doi.org/10.24380/k966-3957, Guevara et al., 2022). The CAMS-REG_v5.1 BAU 2020 gridded emission inventory (https://doi.org/10.24380/eptm-kn40, Kuenen et al., 2022) is distributed as NetCDF (Network Common Data Format) files from the Emissions of atmospheric Compounds of Ancillary Data (ECCAD) system, which will be complemented with access through the ECMWF Atmosphere Data Store (ADS) as soon as this is technically feasible. For review

purposes, ECCAD has set up an anonymous repository where a sample of the emission file can be accessed directly (eccad.aeris-data.fr/essd-surf-emis-cams-reg/).

## 6    Conclusions

We present a dataset of daily sector-, country- and pollutant-dependent emission adjustment factors that allows quantifying the impact of the COVID-19 restrictions on European primary emissions of criteria

pollutants and greenhouse gases for 2020. The dataset was constructed considering changes observed in metrics traditionally used to estimate emissions, such as energy statistics or traffic counts, as well as information derived from new mobility indicators, meteorological data and machine learning techniques. The resulting dataset allows reflecting the heterogeneous impact of COVID-19 restrictions across





countries on air pollutants and greenhouse gases levels for a total of nine anthropogenic activity sectors, including road transport, energy industry, manufacturing industry, residential and commercial combustion, aviation, shipping, off-road transport, use of solvents and fugitive emissions from transportation and distribution of fossil fuels. To the authors knowledge, this is currently the most comprehensive and complete European dataset for inferring changes in primary emissions derived from the COVID-19 restrictions. It is worth noting the intercomparison exercise performed between observed changes in traffic activity derived from governmental traffic flow data and from the Google mobility trends, the latter being widely used in the current literature. Results indicate large deviations between novel Google mobility and traditional traffic flow data, which in the present work were reduced by constructing a set of adjustment factors to better reflect changes in emissions from light-duty and heavy-duty vehicles.

We combined the resulting COVID-19 adjustment factors with the European CAMS-REG gridded (0.1 x 0.05 deg) emission inventory for 2020 following a business-as-usual (BAU) scenario, to spatially and temporally quantify reductions in emissions from both criteria pollutants and greenhouse gases. The main findings and conclusions are as follows:

- The largest decrease in European emissions in 2020 attributed to the COVID-19 lockdown measures were found for $NO_x$ (-10.5%) and $CO_2$ fossil fuel (-7.8%) emissions. For these two pollutants, the most pronounced drop in emissions was found during April (-32.8% and -25.5%) when the mobility restrictions were at their maximum.

- By the end of summer, the effect of COVID-19 measures on emissions diminished as lockdown restrictions relaxed, and emissions remained at values of -4.8% and -3.9% below business-as-usual levels for NOx and $CO_2\_ff$.

- The emission reductions observed during the second epidemic wave (October, November and December) were between three and four times lower than those occurred during the Spring lockdowns, up to -10.5% for NOx and -6.5% for $CO_2$ fossil fuel, since mobility restrictions were





generally softer and only had to be toughened in those countries affected by increasing rates of transmission such as France, Germany or the UK.

- Lower drops in emissions were found for $PM_{10}$ and $PM_{2.5}$ (-3.0% and -2.1%), as these were modulated by residential combustion activities, which slightly increased during the lockdowns. $NH_3$ and $CH_4$ emissions, which are mainly linked to agricultural activities, were practically unaffected by COVID-19 restrictions (-0.9% and -0.2%).

- At the country level, the largest relative emission declines were reported for Italy, UK, Spain and France: between -15.1% and -13.5% for $NO_x$ and -11.4% and -10.4% for $CO_2$ fossil fuel emissions.

- At the sectoral level, the largest emission declines were found for aviation (between -51 and -56%) and road transport (between -15.5% and -18.8%). A drop of similar intensity was observed for both sectors at the beginning of the pandemic. However, while aviation emissions remained almost unchanged, road transport started to gradually recover during late April and the beginning of May, and they reached values of around -5% below BAU by the end of September. A decrease ~50% lower than in April was observed during the second epidemic wave.

- For the other stationary combustion activities, the pollutants that are mainly related to residential wood combustion processes (i.e., $PM_{10}$, $PM_{2.5}$, $NH_3$, NMVOC, CO, $CO_2\_bf$ and $CH_4$) experienced a slight increase (between 1.1% and 1.7%), while the rest of pollutants (i.e., $NOx$, $SO_2$ and $CO_2\_ff$) showed a modest decrease (between -0.4% and -2.9%). Similarly, for the manufacturing industry a heterogenous impact of the COVID-19 restrictions is observed across pollutants: a lower reduction is observed for NMVOC and $NH_3$ (between -2.8% and -3.5%) when compared to the other pollutants (between -6.8% and -7.2%) as these two are mostly driven by processes occurring in the food/beverage and chemistry industries, which were considered to be essential during the Spring lockdowns. Emissions from this sector quickly picked up again, approaching its pre-pandemic levels of activity during November. Unlike other sectors such as road transport, the manufacturing industry remained almost unaffected by the measures implemented to curb the second wave of infections





- The largest contributions to the EU27 + UK decrease in emissions comes from the road transport sector for the majority of pollutants: up to 70.5% for NOx emissions.

- In terms of spatial analysis, the largest emissions reductions occurred in urban areas and main interurban roads. Isolated and significant emission drops were also observed where large point sources are located. The decline in NOx urban emissions was in average 3.4 times larger than the one obtained for NMVOC (-11.3% versus -3.3%).

## 6.1    Limitations of the dataset

The collection of COVID-19 emission adjustment factors and the CAMS-REG_v5.1 2020 BAU inventory have been produced using state-of-the-art information and methods in support of air quality modelling studies. There exist, however, some limitations associated with the current version of the datasets that users should be aware of:

- The emission adjustment factors do not take into account potential variations within each country. This includes, for instance, the heterogeneous lockdown easing process across the different administration units, which may entail heterogeneous recovery rates of the road transport emissions. Similarly, within sea-regions the drop in passenger ship movements (e.g., cruise) during 2020 compared to 2019 has been significantly larger than the one observed for cargo ship
movements. This fact implies that the COVID-19 impact on shipping emissions may not only vary per sea region, but also (and more significantly) per ship route.
- The current factors do not consider the potential impact on NMVOC emissions from residential use of solvents derived from the increase on the consumption of the so-called pandemic products such as hand sanitizers. In the present work, we only assessed the impact of COVID-19 on
industrial use of solvents due to the lack of more detailed data.
- The methodology developed to calculate CAMS-REG gridded emissions for recent years has been validated against reported emissions and shows good results for most sectors and pollutants. The activity data captures a lot of the year-to-year variability, except sudden changes due to, e.g., the closing of a power plant. However, to get a BAU inventory we altered the methodology by





040         ignoring all activity data that may see an impact from the COVID-19 restrictions. This means that, besides the COVID-19 impact, part of the normal year-to-year variability may also be lacking.

## 6.2    Future perspective

Despite the aforementioned limitations, we believe that these emission datasets will allow to refine our understanding of concentration changes observed by satellite and in situ observations, and pinpoint the

effect of COVID-19-related measures more precisely. It will also allow accurate estimates of how far these temporary concentration changes have improved air quality and lowered the related morbidity and mortality. The results reported by Badia et al., (2021), Barré et al., (2021), Guevara et al., (2021) and Schneider et al. (2021), among others, which have made use of previous versions of the emission adjustment factors dataset presented in this work, are proof of that. In this sense, future works will include

using the resulting emission datasets to extend current air quality simulations to the whole year 2020. We also expect to perform intercomparisons of our estimated emission changes against results reported by other existing datasets (e.g., Doumbia et al., 2021; Liu et al., 2020; Forster et al., 2020) as well as the 2020 national official reported emissions when available. This intercomparison exercise will allow us, on the one hand, to assess the consistency between emission results and, on the other hand, to compare and

contrast emission results derived from traditional estimation methodologies used for official reporting against new methods that make use of mobility data sets and other types of near-real time information. Finally, future works will also investigate the potential temporal extension of the emission adjustments factors to 2021, to include the effect of the restrictions and hard lockdowns that were still in place in specific countries such as UK or Germany during Winter time and that may have an effect on main modes

of transport including road traffic or aviation.



# 7    Appendix A

**Table A1: Summary of the European traffic count datasets considered, including: country, source of information, temporal resolution of the traffic counts, vehicles categories included (LDV, Light duty vehicles; HDV, Heavy duty vehicles) and number of observations.**

| Country | Source of information | Temporal resolution | Vehicle categories | Observations |
|---|---|---|---|---|
| Austria | ASFiNAG (2021) | Monthly | LDV / HDV | ~ 275 automatic traffic stations across Austrian road transport network |
| Belgium | FTCC (2021) | Weekly | LDV / HDV | ~ 400 traffic stations distributed over the Flemish Road transport network |
| Denmark | DRD (2021) | Weekly | LDV / HDV | 30 selected stations distributed over Danish Road transport network |
| Estonia | ERA (2021) | Weekly | All | 3 measurement stations representing urban, highway and recreational roads |
| Finland | FTIA (2021) | Daily | All | ~ 500 traffic measuring stations across Finnish road transport network |
| France | CEREMA (2021) | Daily | All | Measurement stations located in the cities of Paris, Toulousse, Nantes, Strasbourg, Bordeaux, Marseille, Lyon and St. Etienne |
| Germany | BASt (2021) | Monthly | LDV / HDV | ~ 800 automatic traffic stations across national and federal German highways |
| Ireland | TII (2021) | Daily | All | ~ 445 automatic traffic stations across Irish road transport network |
| Italy | ANAS (2021) | Monthly | LDV / HDV | ~ 800 automatic traffic count sites across national highways in Italy |
| Luxembourg | MMTP (2021) | Monthly | All | 25 automatic traffic stations across national highways in Luxembourg |
| Netherlands | NWD (2021) | Daily | LDV / HDV | ~ 1600 automatic traffic station from national road network located near the cities of Amsterdam, Roterdam, Eindhoven, Utrecht and The Hague |
| Norway | NPRA (2021) | Daily | All | ~ 720 automatic traffic station located in European and National roads in Norway |
| Poland | Autostrady (2021) | Weekly | LDV / HDV | A-4 motorway section between Katowice and Kraków |
| Portugal | IMT (2021) | Monthly | All | ~ 600 automatic traffic stations across Portuguese national highways |
| Spain | AM (2021), ATM (personal communication) | Daily | All | ~ 60 automatic traffic stations located in the cities of Barcelona and Madrid |
| Sweden | STA (2021) | Weekly | All | 80 automatic traffic stations across the Swedish state road network |
| Switzerland | OFROU (2021) | Daily | LDV / HDV | 10 measurement stations across Swedish national road network |
| United Kingdom | DfT (2021) | Daily | LDV / HDV | ~ 275 automatic traffic stations across British national road network |






## 8 Authors contributions

MG conceived and coordinated the study as well as the development of the COVID-19 emission adjustment factors. J-PJ, EM and LJ provided the estimation of 2019 and 2020 shipping emissions using the STEAM model, which were used for the development of the shipping emission adjustment factors. HP developed the machine learning algorithm for computing business-as-usual electricity demand during 2020. HDvdG, JK and IS developed the CAMS-REG business-as-usual 2020 emission inventory and have provided comments about the work. VHP has provided comments about the work and ensured liaison with wider activities in CAMS related to COVID-19. OJ and CPGP helped conceive the COVID-19 emission adjustment factor dataset and supervised the work. MG prepared the manuscript with contributions from all co-authors.

## 9 Competing interests

The authors declare that they have no conflict of interest.

## 10 Acknowledgements

The research leading to these results has received funding from the Copernicus Atmosphere Monitoring Service (CAMS), which is implemented by the European Centre for Medium-Range Weather Forecasts (ECMWF) on behalf of the European Commission. We acknowledge support from the Ministerio de Ciencia, Innovación y Universidades (MICINN) as part of the BROWNING project RTI2018-099894-B-I00, from the VITALISE project (PID2019-108086RA-I00) funded by the MCIN/AEI /10.13039/501100011033, the MITIGATE project (PID2020-116324RA695 I00 / AEI / 10.13039/501100011033) from the Agencia Estatal de Investigacion (AEI), the AXA Research Fund, and the European Research Council (grant no. 773051, FRAGMENT). This project has also received funding from the European Union's Horizon 2020 research and innovation programme under the Marie Skłodowska-Curie grant agreement H2020-MSCA-COFUND-2016-754433. BSC researchers thankfully acknowledge the computer resources at Marenostrum and the technical support provided by Barcelona Supercomputing Center (RES-AECT-2021-1-0027, RES-AECT-2021-2-0001).



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



**Table 1 Summary of information sources used to compute the emission adjustment factors for each sector.**

| Sector | Description | Sources of information |
|---|---|---|
| GNFR_A | Public power industry | • Electricity demand data: ENTSO-E (2021), SO-UPS (2021), TEIAS (2021), UNEC (2021)<br>• Outdoor temperature: ERA5 reanalysis (C3S, 2017)<br>• Population map: CIESIN (2016) |
| GNFR_B | Manufacturing industry | • Industrial Production Index: Eurostat (2021a), ONS (2021)<br>• Energy balances: Eurostat (2021b)<br>• Oxford COVID-19 Government Response Tracker: Hale et al. (2021) |
| GNFR_C | Other stationary combustion activities | • Google movement trend reports: Google (2021)<br>• Consumption by use for the commercial sectors: IDAE (2018) |
| GNFR_D | Fugitive emissions from fossil fuels | Industrial Production Index: Eurostat (2021a), ONS (2021) |
| GNFR_E | Solvents | Industrial Production Index: Eurostat (2021a), ONS (2021) |
| GNFR_F1, GNFR_F2, GNFR_F3 & GNFR_F4 | Road Transport | • Google movement trend reports: Google LCC (2021)<br>• Traffic count datasets from national transport agencies (see Table A1 for complete list of references) |
| GNFR_G | Shipping | • AIS-based shipping emissions: Jalkanen et al. (2012 and 2016) |
| GNFR_H | Aviation | Airport movement statistics: EUROCONTROL (2021) |
| GNFR_I | Off-road transport | Industrial Production Index: Eurostat (2021a), ONS (2021) |




**Table 2 Subcategories considered for the development of the adjustment factors for each GNFR sector**

| Sector | Subcategories |
|---|---|
| GNFR_B | • GNFR_B1: Manufacture of petroleum refining products<br>• GNFR_B2: Manufacture of pharmaceutical, chemistry, food and beverages products<br>• GNFR_B3: Manufacture of other products (e.g., non-metallic mineral products, basic metals) |
| GNFR_C | • GNFR_C1: Commercial/Institutional stationary combustion activities<br>• GNFR_C2: Residential combustion stationary activities<br>• GNFR_C3: Other combustion stationary activities (Agriculture/Forestry/Fishing) |
| GNFR_D | • GNFR_D1: Fugitive emission from solid fuels: Coal mining and handling<br>• GNFR_D2: Fugitive emissions oil: Refining / storage & venting and flaring<br>• GNFR_D3: Distribution of oil products<br>• GNFR_D4: Other activities not affected by COVID-19 restrictions |
| GNFR_E | • GNFR_E1: Degreasing<br>• GNFR_E2: Printing<br>• GNFR_E3: Other activities not affected by COVID-19 restrictions |
| GNFR_F2<br>GNFR_F4 | • GNFR_F21/GNFR_F41: Passenger cars, light duty vehicles, mopeds and motorcycles<br>• GNFR_F22/GNFR_F42: Heavy duty vehicles and buses |
| GNFR_I | • GNFR_I1: Mobile Combustion in manufacturing industries and construction<br>• GNFR_I2: Other activities not affected by COVID-19 restrictions |



**400** **Table 3 Overview of activity data used for each emission sector and subcategory as defined in Table 1 and Table 2 to derive the BAU 2020 emissions and expected COVID-19 impact**

| Sector/Subcategory | Activity data | COVID-19 |
|---|---|---|
| GNFR_A | Electricity generation (non-renewable)[1] | Yes |
| GNFR_B1 | Refinery througput[2] | Yes |
| GNFR_B2 | Industrial production index (manufacturing)[3] | Yes |
| GNFR_B3 | Industrial production index (manufacturing)[3] | Yes |
| GNFR_C1 | Yearly degree day sum[4] | No |
| GNFR_C2 | Yearly degree day sum[4] | No |
| GNFR_C3 | Yearly degree day sum[4] | No |
| GNFR_D1 | Coal production[2] | Yes |
| GNFR_D2 | Refinery througput[2] | Yes |
| GNFR_D3 | Industrial production index (manufacturing)[3] | Yes |
| GNFR_D4 | Industrial production index (manufacturing)[3] | Yes |
| GNFR_F1, GNFR_F21, | Energy consumption in transport sector[5] | Yes |
| GNFR_K (livestock) | Animal numbers (cattle, swine, sheep, other)[6] | No |
| GNFR_L (Application of manure and fertilizer) | Total nutrient N from agricultural fertilizer use[7] | No |
| GNFR_L (Other) | Utilised agriculture area[8] | No |

[1] ENTSO-E (2021); [2] BP (2020); [3] Eurostat (2021a); [4] C3S (2017); [5] Eurostat (2021c); [6] FAO (2021a); [7] FAO (2021b); [8] Eurostat (2021d)

**405**



**Figure 1: Daily COVID-19 emission adjustment factors computed per GNFR sector and pollutant for selected countries: Germany (DE), Spain (ES), France (FR), the United Kingdom (GB), Italy (IT) and Sweden (SE). For the shipping sectors, adjustment factors are reported for selected sea regions: Atlantic Ocean (ATL), Baltic Sea (BAS), English Channel (ENC), Mediterranean Sea (MED), North Sea (NOS) and Norwegian Sea (NWS). For the GNFR sectors A (public power), H (aviation) and G (shipping) the constructed adjustment factors are the same for all species. Adjustment factors are reported for the period 21 February to 31 December 2020.**




**Figure 2 Evolution of the average monthly Industrial Production Index (IPI) values (January 2019 to December 2020) for four manufacturing categories: Group01 (petroleum refining products), Group02 (pharmaceutical, chemistry, food and beverages products), Group03 (non-metallic mineral products, basic metals, paper and machinery and equipment) and All (general index for all the manufacturing industry). Results are presented for Germany (DE), Spain (ES), Italy (IT) and the United Kingdom (GB). The IPI values are seasonally and calendar adjusted data (Eurostat, 2021a)**







**Figure 3 Average contribution of each GNFR subcategory (see definitions in Table 2) to total annual emissions for selected pollutants per country (EU27 + UK) for year 2020.**




**Figure 4 Comparison of traffic movement trends derived from Google COVID-19 Community Mobility Reports (Google LLC, 2021) and measured traffic counts for selected countries (see Table A1 for references), the latter one being distinguished by type of vehicle (i.e. heavy duty vehicles, HDV; light duty vehicles and cars, LDV), for the period 21 February to 31 December 2020.**







**Figure 5 Comparison between averaged monthly adjustment factors for road traffic reported by Google COVID-19 Community Mobility Reports (Google LLC, 2021) and light duty vehicles and cars (LDV) measured-based trends per country (see Table A1 for references).**



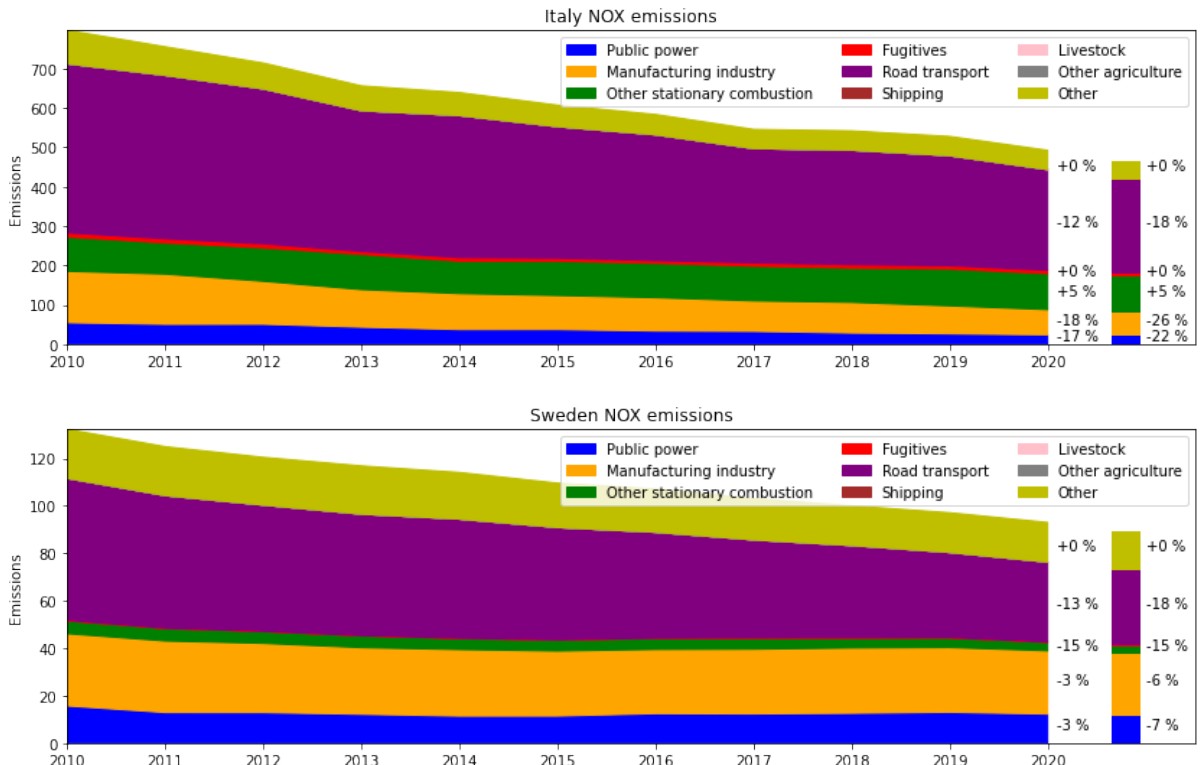

**435** **Figure 6 Time series of NOₓ emissions [kg/year] for Italy and Sweden. Up to 2018 official reported emissions are used. For 2019 and BAU 2020 emissions are estimated. For 2020 a second estimate is made (separate bar on the right) that includes AD affected by COVID-19. Percentages refer to the difference compared to 2018.**


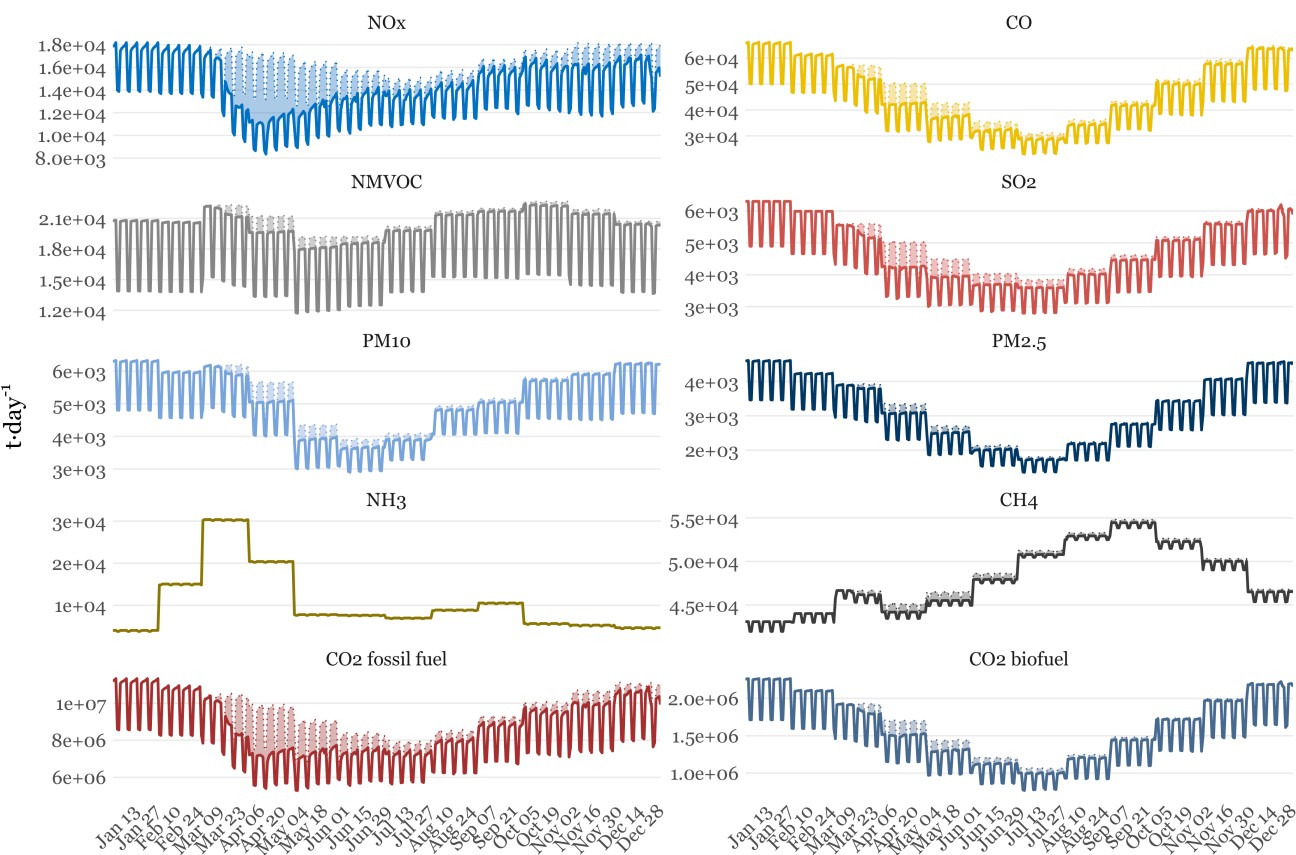

**Figure 7 Daily emissions [t·day⁻¹] by pollutant computed for the 2020 business-as-usual (BAU) (dotted lines) and COVID-19 (solid lines) scenarios between January 1ˢᵗ and December 31ˢᵗ 2020 for EU27 + UK. The areas highlighted between the two lines represent the emission differences between scenarios.**






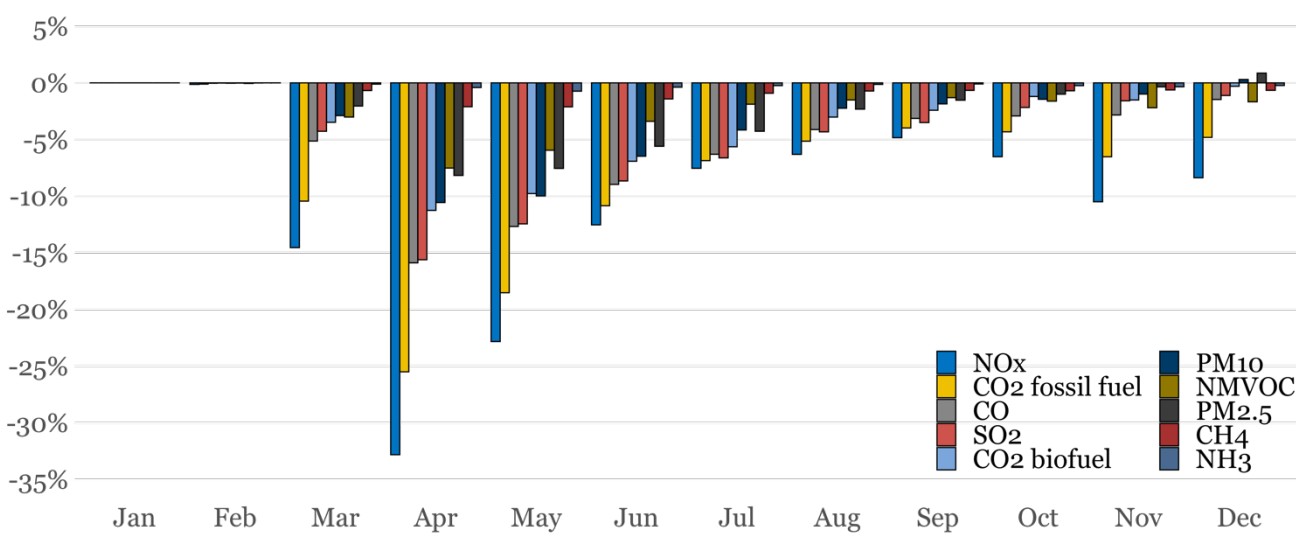

**Figure 8 Relative decline in monthly emissions [%] by pollutant between January and December 2020 for EU27 + UK.**






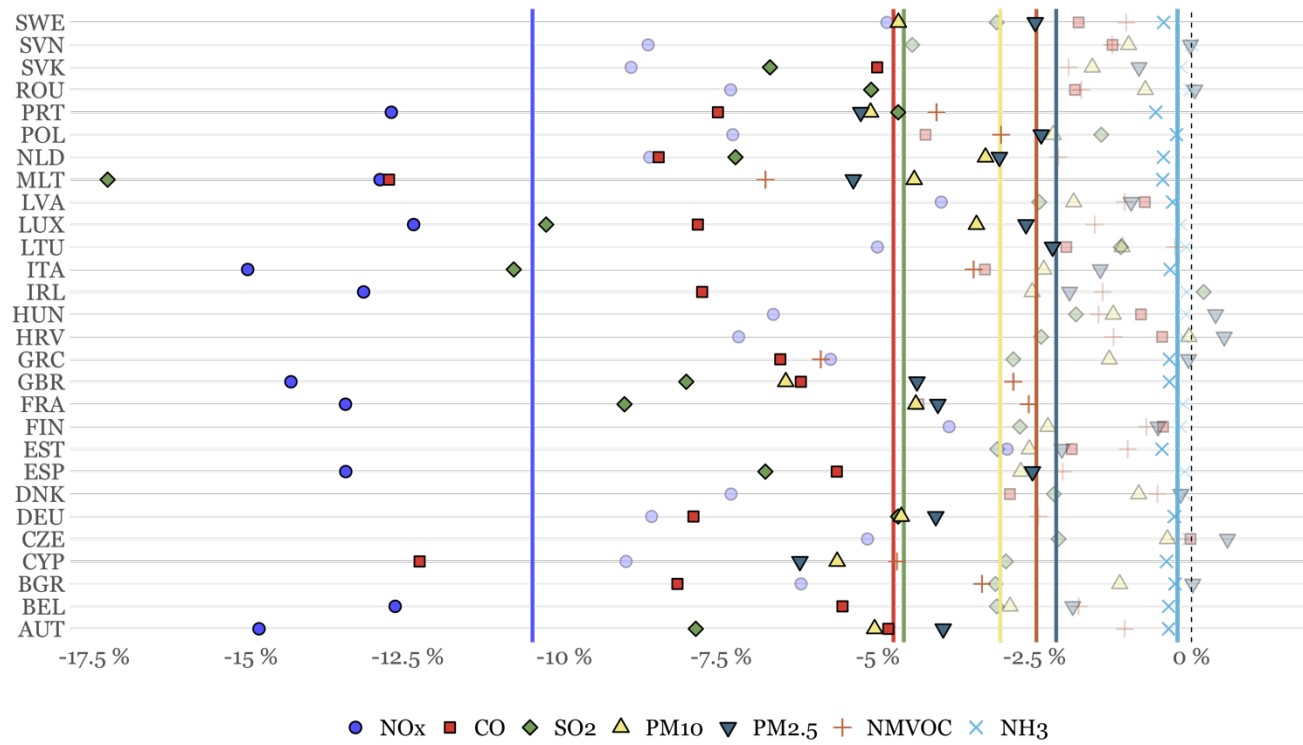

**Figure 9 Relative decline in emissions of criteria pollutants [%] per species and country in 2020. The vertical lines indicate the average relative declines at the EU27 + UK level. Non-shaded marks highlight those countries/species for which reductions are larger than the ones computed at the EU27 + UK level.**






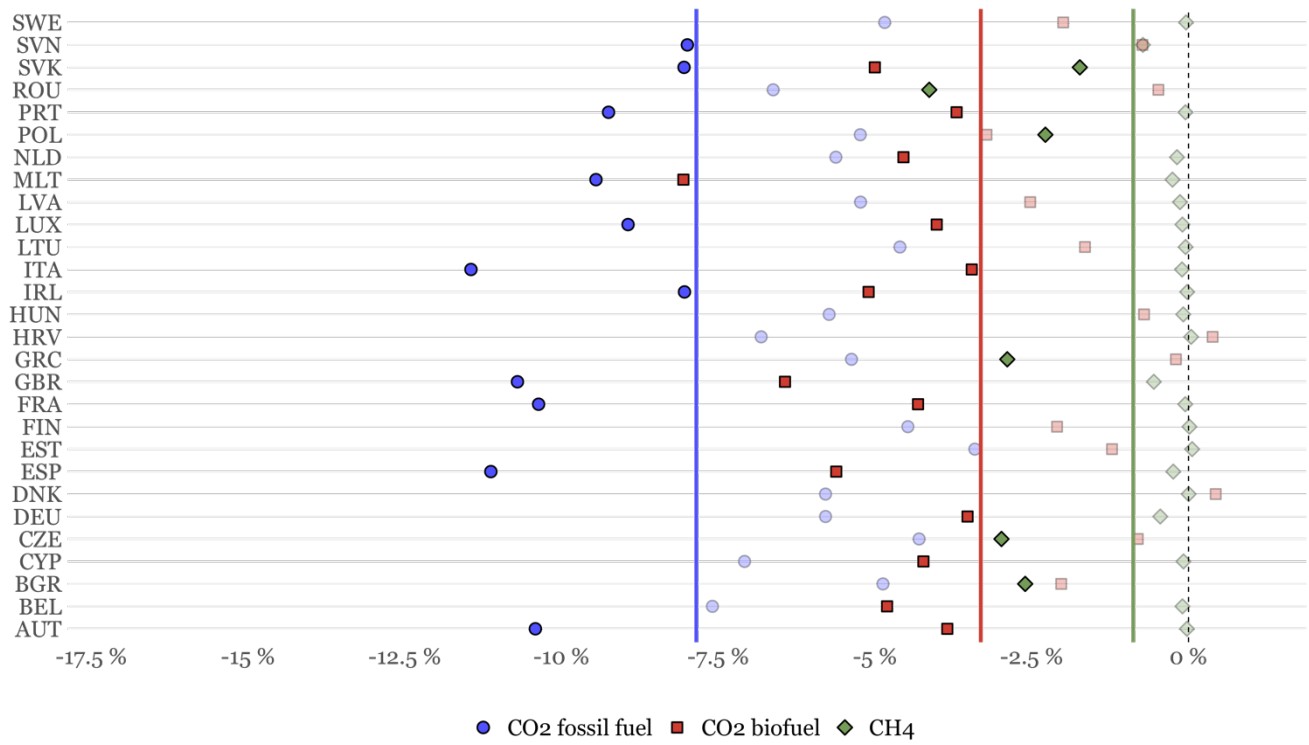

**Figure 10 Same as Figure 9 for greenhouse gases**



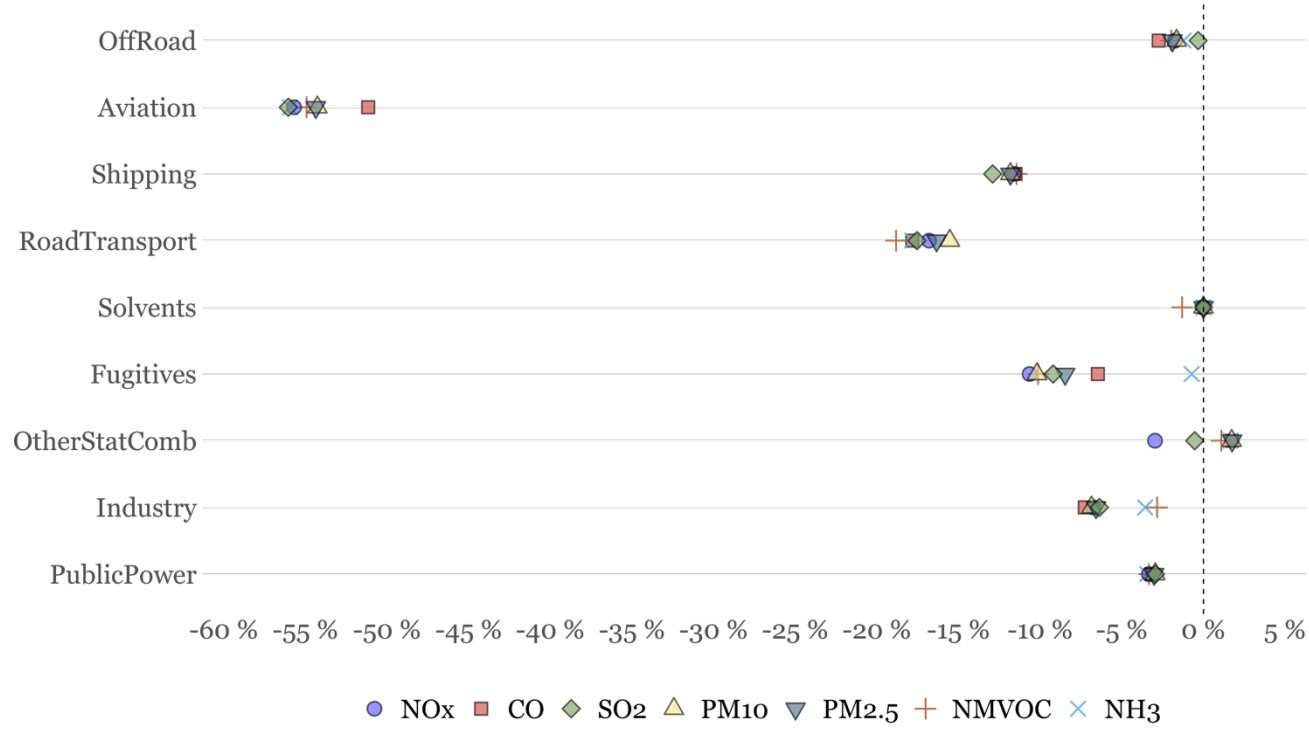


**Figure 11 Relative decline in emissions of criteria pollutants [%] by sector and species between January 1st and December 31st 2020 for EU27 + UK. For the shipping sector the relative differences consider both inland and sea shipping sectors.**





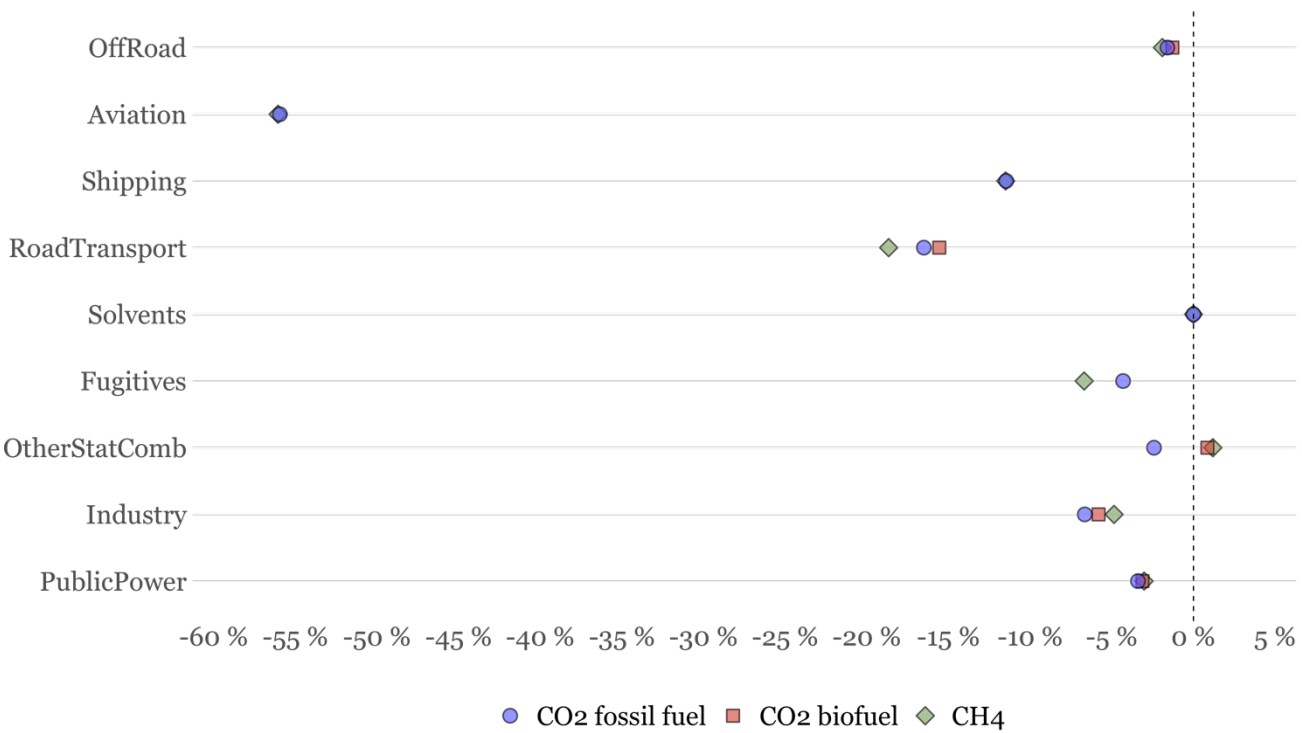

**Figure 12 Same as Figure 11 for greenhouse gases. Note that for Aviation, shipping, use of solvents and fugitives no emissions are reported for CO₂ biofuel.**





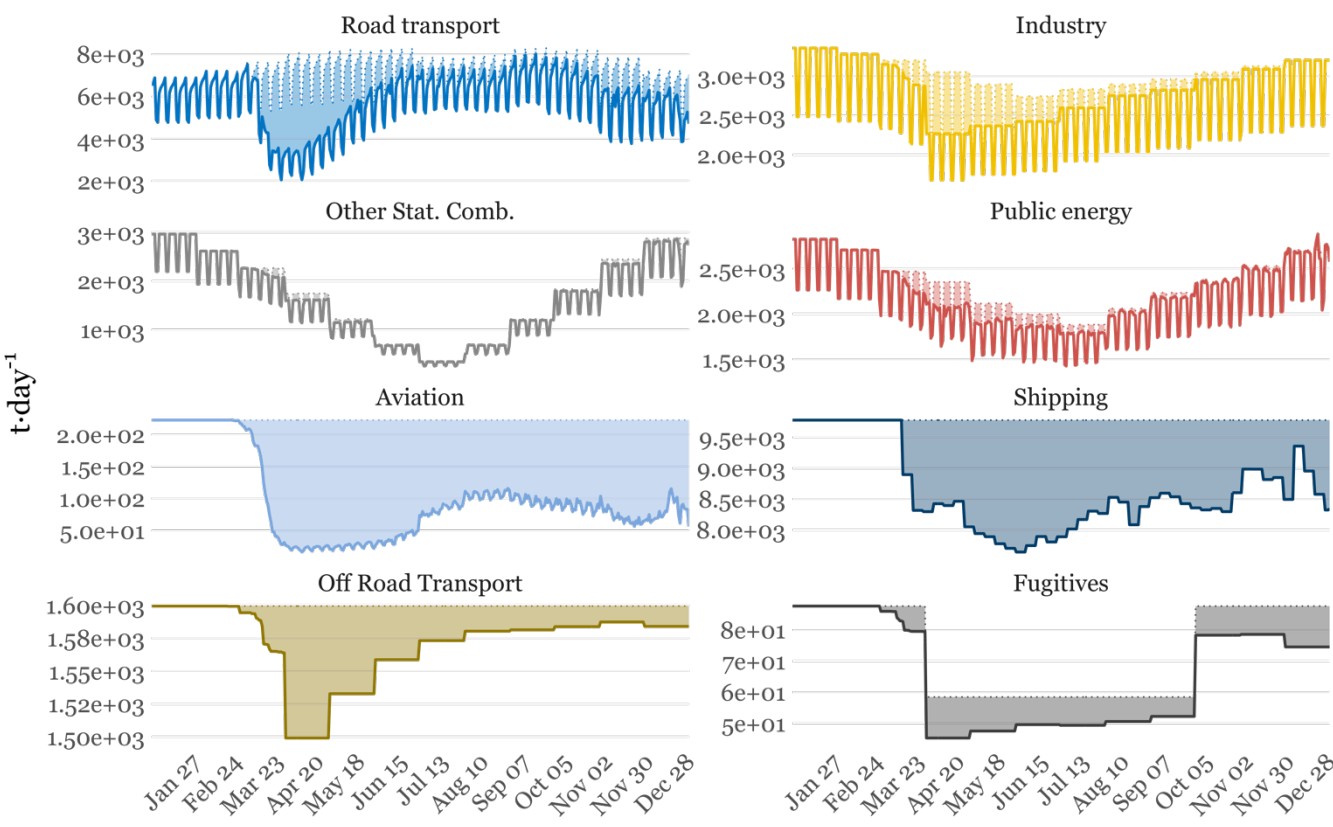

**Figure 13 Daily NOx emissions [t·day-1] by sector computed for the 2020 business-as-usual (dotted line) and COVID-19 (solid line) scenarios between January 1st and December 31st 2020 for EU27 + UK. For the shipping sector the relative differences consider both inland and sea shipping sectors. The areas highlighted between the two lines represent the emission differences between the two scenarios.**





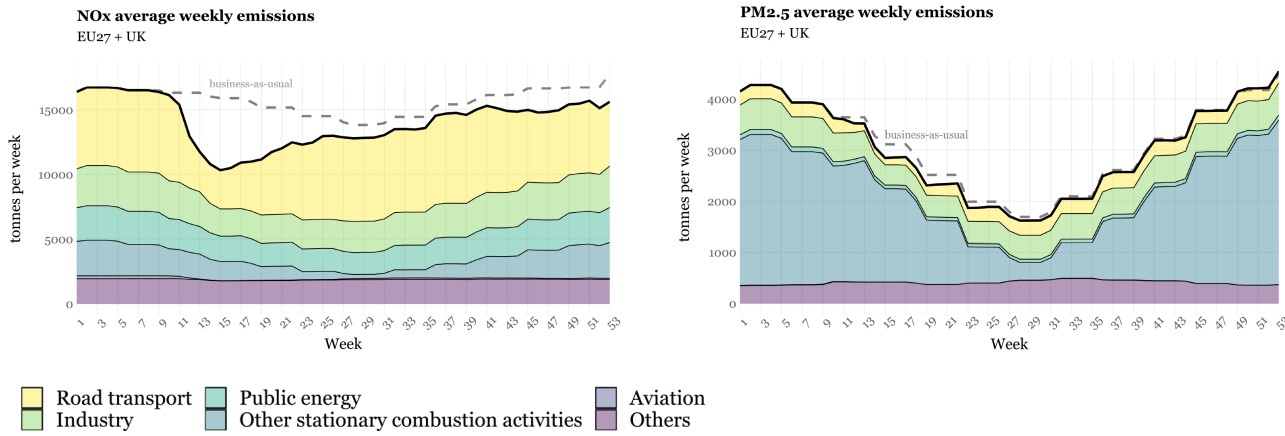

**Figure 14 Stacked area charts representing the evolution of the average weekly emissions of NO$_x$ (left) and PM$_{2.5}$ (right) per pollutant sector in EU27 + UK during the COVID-19 pandemic per pollutant sector.**







**Figure 15** Map of the absolute cumulative $NO_x$ emission decline [kg·cell$^{-1}$] in 2020 as compared to the business-as-usual scenario. Gridded emission changes are reported at a resolution of 0.1 x 0.05 degrees. Administrative boundaries are derived from the Micro World Data Bank (MWDB2, 2011). (Top), Average (dark red) and 5$^{th}$ and 95$^{th}$ percentiles (light blue shading) of the relative changes [%] in gridded NOx emissions in Italy (bottom-left) and Germany (bottom-right) for the period 1 January to 31 December 2020.





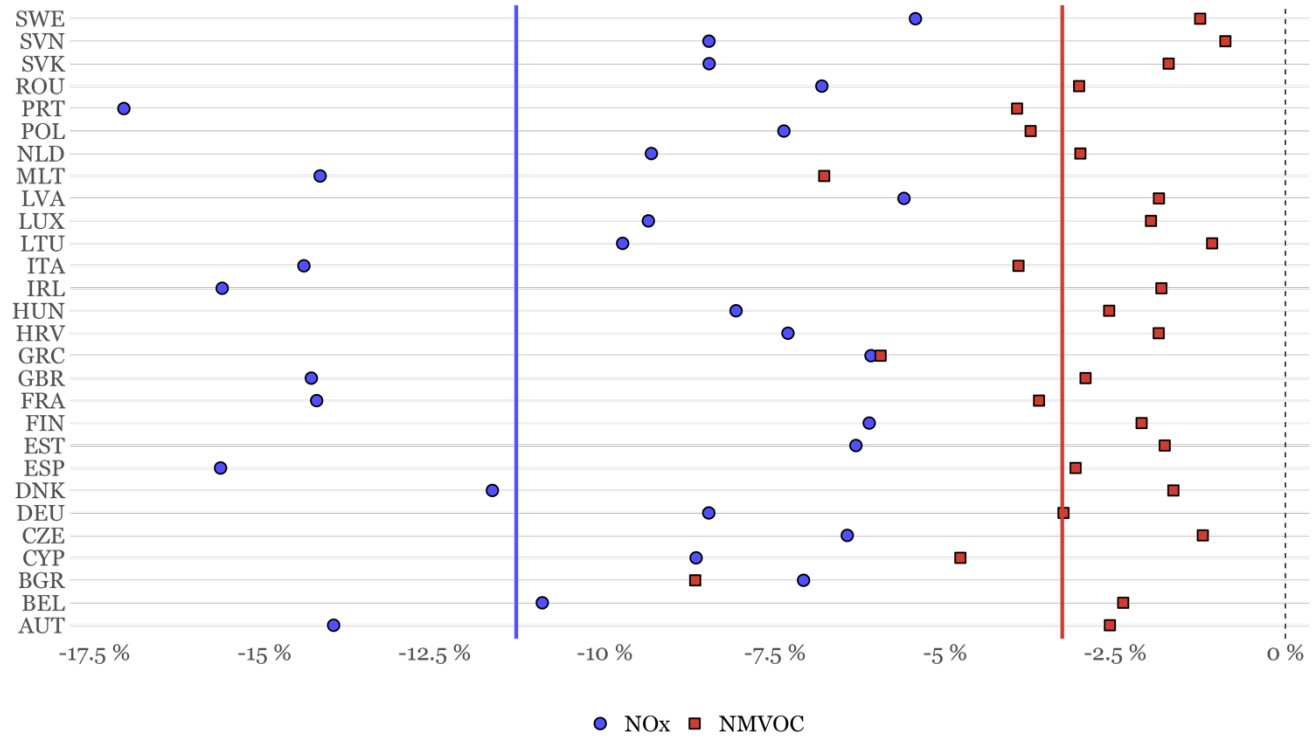


**Figure 16 Relative NO$_x$ and NMVOC emission declines [%] per country occurring in high-density urban areas between January 1$^{st}$ and December 31$^{st}$ 2020. High-density urban areas were defined according to the Global Human Settlement Layer (GHSL) project (Pesaresi et al., 2019). The blue and red solid lines represent the average NO$_x$ and NMVOC emission declines at the EU27 + UK level, respectively.**