# Peer review of "European primary emissions of criteria pollutants and greenhouse gases in 2020 modulated by the COVID-19 pandemic disruptions"

_Earth System Science Data, 2022_

## Referee Comment (RC1)

**General comments:**

This paper describes a dataset of temporal daily adjustment factors for the year 2020 due to the Pandemic outbreak which can be applied to emission inventories of primary pollutants. This dataset covers the whole European country and is based on several proxy combined together to provide a new improvement of a previous set of data released in 2021. This kind of dataset is very useful to adapt the emissions to this peculiar period and provide a framework of continuous adaptation of emissions along this sanitary crisis.

I think the paper is suitable for publication after the consideration of the following comments.

**Specific comments:**

1) My major comment regards the proxy used for the adjustment of GNFR_C emissions. I think a proxy related to energy consumptions of households (electricity or fuels) could be more relevant than google mobility data. Could the authors discuss this point and did they try to find this proxy in European databases?
2) The paper lacks a comment on the emissions regarding the agricultural sector. In most studies related to the COVID outbreak the implicit assumption is that emissions from agriculture (mainly ammonia) was not affected. It could be interesting to prove this with satellite data or find a reference paper to add (see references here below for suggestions).
3) To help the reader, I would prefer to have first the section describing the BAU that is the reference and after the description of the COVID19 case. It should be more logical.
4) It is not clear how the authors handle the meteorological effect on emission to build the BAU reference so that the adjustment just reflect the restrictions and lockdowns due to the pandemic. This is an important point to clarify for modellers to help them to correctly use these daily factors.
5) In some countries there are some hourly variation of traffic counts that could be added as an update to get some flavours of the hourly variations we could extrapolate at the European level (See CEREMA web site in France).
6) In the discussion you could add something for FR with a rebound of NOx emissions in August with higher emissions compared to BAU in Adelaide et al. (2021)
7) In the limitations I suggest the authors to elaborate more on the spatial variation within the country with probably a decrease of emissions in very urbanized cities impacted by the COVID (Paris for instance) and in the countryside probably an increase of emissions (particularly for the PM from wood burning). This would be due to an *exodus* from city centres toward remote areas during the sanitary crisis.

I would suggest to add these missing references:

- *Adélaïde, L., Medina, S., Wagner, V., de Crouy-Chanel, P., Real, E., Colette, A., Couvidat, F., Bessagnet, B., Alter, M., Durou, A., Host, S., Hulin, M., Corso, M., Pascal, M., 2021. Covid-19 Lockdown in Spring 2020 in France Provided Unexpected Opportunity to Assess Health Impacts of Falls in Air Pollution. Front. Sustain. Cities 3, 643821. https://doi.org/10.3389/frsc.2021.643821.*

In this reference there is an annex with a methodology to calculate adjustment factors with a tentative retrieve of an indicator of house hold consumption to address GNFR_C residential emissions.

- This web site: https://www.citepa.org/fr/barometre/
Citepa as an official organism providing emissions release a monthly evolution of emissions.

- *Zhang, Y., Liu, X., Fang, Y., Liu, D., Tang, A., Collett, J.L., 2021. Atmospheric Ammonia in Beijing during the COVID-19 Outbreak: Concentrations, Sources, and Implications. Environ. Sci. Technol. Lett. 8, 32–38. https://doi.org/10.1021/acs.estlett.0c00756.*

  *Lovarelli, D., Conti, C., Finzi, A., Bacenetti, J., Guarino, M., 2020. Describing the trend of ammonia, particulate matter and nitrogen oxides: The role of livestock activities in northern Italy during Covid-19 quarantine. Environmental Research 191, 110048. https://doi.org/10.1016/j.envres.2020.110048*

  These studies could be commented to discuss NH3 emissions.

- *Gkatzelis, G.I., Gilman, J.B., Brown, S.S., Eskes, H., Gomes, A.R., Lange, A.C., McDonald, B.C., Peischl, J., Petzold, A., Thompson, C.R., Kiendler-Scharr, A., 2021. The global impacts of COVID-19 lockdowns on urban air pollution. Elementa: Science of the Anthropocene 9, 00176. https://doi.org/10.1525/elementa.2021.00176*

  In the introduction, I would cite this study that is a good review on the impact of COVID restrictions on air quality.

**Technical comments**

Very few minor comments, the paper is very well written, just chose either American or UK English. To help the reader I would add a list of acronyms and abbreviations.

Through the publication, please correct **Le Quéré** with two accents (when necessary).

Line 156 : heterogen**e**ous impact (same L 839 and L 1005)

Line 239: This sentence could be improved "Italy is where the recovery is more pronounced, reaching emissions above the BAU during August"

Line 394: I would write "Until the end of August, most"

---

## Author Response (AR1)

We would like to thank the reviewers for their positive and constructive feedback, which helped improve the quality of the paper The review comments have been helpful in pointing out parts that required further improvements. Below we address specific issues mentioned by the reviewers point by point. The manuscript has been updated accordingly.

**Anonymous Referee #1**

This paper describes a dataset of temporal daily adjustment factors for the year 2020 due to the Pandemic outbreak which can be applied to emission inventories of primary pollutants. This dataset covers the whole European country and is based on several proxy combined together to provide a new improvement of a previous set of data released in 2021. This kind of dataset is very useful to adapt the emissions to this peculiar period and provide a framework of continuous adaptation of emissions along this sanitary crisis. I think the paper is suitable for publication after the consideration of the following comments.

Specific comments:

1) My major comment regards the proxy used for the adjustment of GNFR_C emissions. I think a proxy related to energy consumptions of households (electricity or fuels) could be more relevant than google mobility data. Could the authors discuss this point and did they try to find this proxy in European databases?

The GNFR_C sector includes emissions from stationary combustion activities related to multiple sources, including the commercial and institutional sector, the residential sector and other stationary sectors such as agriculture, forestry and fishing. As pointed out in the manuscript, COVID-19 restrictions impacted emissions from these three sources in a completely heterogeneous way: 1) reductions in the commercial/institutional sector due to the closure of schools, universities, public buildings, restaurants, and other non-essential businesses, 2) increases in the residential sector due to the required household confinement and 3) negligible changes in the agriculture/forestry/fishing sectors as these activities were considered to be essential.

Considering all of the above, and as highlighted by the reviewer, a proper estimation of the impact of COVID-19 restrictions on this sector would require the use of energy proxies that distinguish between what occurred in the residential and commercial/institutional sectors. However, and to the author's knowledge, there is currently no open access dataset that provides near-real time and high temporal resolution information on European energy consumption for the residential and commercial sectors separately. The closest dataset to meet these characteristics is the ENTSO-G transparency platform (https://transparency.entsog.eu/), which reports data on EU daily natural gas flows toward distribution and final consumption. However, there are several limitations that prevent the use of this dataset, namely: 1) the data does not separate commercial/public and residential buildings and 2) the data is only available for a limited number of EU countries. There are other national databases that face similar problems, such as GRTgaz (https://www.smart.grtgaz.com/en/consommation/GRTgaz), which provides daily consumption of natural gas in France of industrial sites and on public network (without distinguishing between commercial/institutional and residential sectors).

Taking into account the lack of information on energy statistics, we decided to make use of the Google mobility data, as it reports changes in daily movement trends across different categories of places, including commercial/institutional places (i.e., groceries and pharmacies, retail and recreation and workplaces) and residential areas. Knowing that changes in movements do not necessarily represent changes in the energy consumption, the original google mobility data was scaled making use of energy consumption statistics available from current literature. The proxies considered to scale the data are from specific locations (UK smart meter data and Spanish commercial/institutional energy statistics) and may not reflect the changes occurred in all European countries. However, one of the advantages of such approach is its consistency across countries.

We have highlighted the limitation of available EU energy consumption statistics and of our approach in the revised version of the manuscript as follows:

"Adjustment factors for the residential and commercial stationary combustion sectors were derived from Google mobility statistics, which may not necessarily represent changes in the energy consumption of these two sources. However, we could not find any open access dataset that provides near-real time and high temporal resolution information on European energy consumption for the residential and commercial sectors separately. The closest dataset meeting these characteristics is the ENTSO-G transparency platform (https://transparency.entsog.eu/, last accessed March 2022), which reports data on EU daily natural gas flows toward distribution and final consumption. However, the data does not separate commercial/public and residential buildings and it is only available for a limited number of EU countries. There are other national databases that face similar problems, such as GRTgaz (https://www.smart.grtgaz.com/en/consommation/GRTgaz, last accessed March 2022), which provides daily consumption of natural gas by industrial sites and the public network in France, without distinguishing between commercial/institutional and residential sectors."

2) The paper lacks a comment on the emissions regarding the agricultural sector. In most studies related to the COVID outbreak the implicit assumption is that emissions from agriculture (mainly ammonia) was not affected. It could be interesting to prove this with satellite data or find a reference paper to add (see references here below for suggestions).

Following the reviewer's suggestion, we included the following comment on the emissions from the agricultural sector:

"Agricultural emissions (GNFR_K for livestock and GNFR_L for other activities including use of fertilizers and agricultural waste burning) were assumed to remain unaffected by the COVID-19 restrictions, as their activities were considered to be essential during lockdown periods. This assumption is consistent with the surface measurement-based results reported by Lovarelli et al. (2020) and Zhang et al. (2021) as well as the results published by Elleby et al. (2020), which indicate that COVID-19 implied a reduction of direct GHGs from agriculture of only about 1% at the global scale."

3) To help the reader, I would prefer to have first the section describing the BAU that is the reference and after the description of the COVID19 case. It should be more logical.

Following the reviewer's suggestion, we have rearranged the sections of the manuscript so that the description of the BAU emission inventory (new Section 2) appears before the description of the COVID-19 emission adjustment factors (new Section 3). References to these Sections and associated numbering have been updated in the manuscript accordingly. Also, the numbering of the figures has been updated accordingly.

4) It is not clear how the authors handle the meteorological effect on emission to build the BAU reference so that the adjustment just reflects the restrictions and lockdowns due to the pandemic. This is an important point to clarify for modellers to help them to correctly use these daily factors.

Changes in meteorology affect the emissions related to heating of buildings, which is part of GNFR_C (other stationary combustion). To estimate recent year emissions from this sector the annual heating degree day sum is used as measure of activity. Since changes in meteorology are independent from COVID-19 we use the heating degree day sum for the BAU as well. Although some other sectors may be influenced by meteorology, especially in their timing, this will play a minor role in estimating annual emissions. The constructed daily COVID-19 adjustment factors are not corrected for the meteorology to avoid a double counting issue. Therefore, the changes reflected in the adjustment factors are only related to changes in the activity, but not in the meteorology. We have clarified this point in the revised version of the manuscript as follows:

"Note that for the other stationary combustion activities (GNFR_C), which includes emissions related to heating of buildings, the annual heating degree day sum is used as measure of the AD to derive 2020 BAU emissions. Thus, we can isolate the impact of 2020 temperatures, which were above the 1981–2010 average across all of Europe (C3S, 2021) and generally reduced the use of fuel for space-heating purposes, from the impact of COVID-19 stay-home orders, which increased the time people spent at home and are considered through the adjustment factors presented in Sect. 3.1.3."

Copernicus Climate Change Service (C3S): Climate bulletin. European State of the Climate in 2020: Temperature. Available at: https://climate.copernicus.eu/esotc/2020/temperature, (last accessed, March 2022) 2021.

5) In some countries there are some hourly variation of traffic counts that could be added as an update to get some flavours of the hourly variations we could extrapolate at the European level (See CEREMA web site in France).

We used the raw measured hourly traffic counts provided by AM (2021) for the city of Madrid to perform a preliminary investigation of the impact of COVID-19 restrictions on the diurnal variation of road traffic activity. The figure below shows the hourly variation of traffic activity during weekdays (left) and Saturdays (right) as a function of the week (from week 4 to week 31). Two groups of profiles with similar behaviours are clearly identified in each case. For weekdays, during the weeks before and after the lockdown (represented in blue and yellow) a maximum level of traffic activity is reached in the morning (between 08:00 and 09:00h LT) that largely remains for the rest of the day-time and through part of the night (until approximately 19:00h-20:00h LT). On the other hand, during the weeks affected by the lockdown (represented in green) two pronounced peaks appear in the early morning (between 08:00 and 09:00h LT) and in the afternoon (between 15:00 and 16:00h LT), the traffic activity being significantly reduced in the evening. This shift in the diurnal distribution of emissions is most likely due to a combination of two factors: (i) the decrease of work-related trips due to the confinement and (ii) the increase of urban freight transport triggered by an increased e-commerce activity. A significant change in the diurnal pattern of traffic activity is also observed for Saturdays, which is triggered by the decrease of nightlife activity during the lockdown. The relatively high traffic activity levels observed during night time under normal conditions are significantly reduced during the lockdown weeks. Moreover, an early morning peak is registered during the lockdown weeks, which could be also related to the increase of e-commerce activity and associated urban freight transport.

[Figure]

**Figure 1. Hourly temporal profiles derived from measured-traffic counts in Madrid city for the year 2020 (AM, 2021) for weekdays (left) and Saturdays (right) discriminated by week of the year.**

In this work we quantified the impact of COVID-19 restrictions on emissions at the daily scale, leaving the potential effects on the emission diurnal patterns out of the scope. The results showed above can be seen as a preliminary assessment. However, more datasets should be compiled and analysed in order to extrapolate robust conclusions and recommendations.

A comment regarding the analysis of these results and the need to further investigate on this topic has been included in the Future perspective section (Section 6.2) of the revised manuscript. The Figure above has been included as part of the revised supplementary material.

"We quantified the impact of COVID-19 restrictions on emissions at the daily scale. A preliminary assessment of the impact upon the hourly variations in road traffic activity in Madrid city indicates a significant shift in the diurnal cycle during weekdays and weekends (Figure S8). Such shift was likely driven by a decrease in work-related trips and nightlife activity, along with an increase in e-commerce activity and associated urban freight transport during the confinement. Future studies may elucidate how hourly emissions changed during lockdown periods and more importantly to what extent these patterns persisted after the easing of the restrictions."

6) In the discussion you could add something for FR with a rebound of NOx emissions in August with higher emissions compared to BAU in Adelaide et al. (2021)

The spatial analysis section of the manuscript already presents two examples of rebound of NOx emissions during August (i.e., Italy and Germany). We have added a mention to France in the text as follows:

"During summer the range of relative changes becomes much lower, with emissions ranging between -10% below and 10% above BAU levels, as mobility restrictions were lifted and traffic activity reached values above BAU levels due to the increase of domestic tourism. This was also observed in France."

7) In the limitations I suggest the authors to elaborate more on the spatial variation within the country with probably a decrease of emissions in very urbanized cities impacted by the COVID (Paris for instance) and in the countryside probably an increase of emissions (particularly for the PM from wood burning). This would be due to an exodus from city centres toward remote areas during the sanitary crisis.

This is indeed a very good point. We have added the following sentence to include this aspect in terms of limitations of the study:

"Last but not least, variations in residential combustion emissions have probably been heterogeneous within countries due to an exodus from city centres toward rural areas during the sanitary crisis. This reallocation may have caused, on the one hand, a decrease of emissions in very urbanized cities impacted by the COVID-19 and, on the other, increases in the countryside, particularly in PM from wood burning activities."

I would suggest to add these missing references:

- Adelaide, L., Medina, S., Wagner, V., de Crouy-Chanel, P., Real, E., Colette, A., Couvidat, F., Bessagnet, B., Alter, M., Durou, A., Host, S., Hulin, M., Corso, M., Pascal, M., 2021. Covid-19 Lockdown in Spring 2020 in France Provided Unexpected Opportunity to Assess Health Impacts of Falls in Air Pollution. Front. Sustain. Cities 3, 643821. https://doi.org/10.3389/frsc.2021.643821.

In this reference there is an annex with a methodology to calculate adjustment factors with a tentative retrieve of an indicator of house hold consumption to address GNFR_C residential emissions.

- This web site: https://www.citepa.org/fr/barometre/

Citepa as an official organism providing emissions release a monthly evolution of emissions.

The Adélaïde et al. (2021) reference has been added in the introduction section, when presenting the current COVID-19 emission works performed in Europe:

"Adélaïde et al. (2021) constructed an emission dataset for France covering strict lockdown and gradual lifting periods (i.e., March to June 2020) using as a basis de adjustment factors from Guevara et al. (2021) together with finer calculations of emission variations by region for road traffic and a first estimate for the residential sector. Information on number of vehicles on the road and household electricity consumption was used to compute the variation of emissions for these two sectors."

Zhang, Y., Liu, X., Fang, Y., Liu, D., Tang, A., Collett, J.L., 2021. Atmospheric Ammonia in Beijing during the COVID-19 Outbreak: Concentrations, Sources, and Implications. Environ. Sci. Technol. Lett. 8, 32–38. https://doi.org/10.1021/acs.estlett.0c00756.

Lovarelli, D., Conti, C., Finzi, A., Bacenetti, J., Guarino, M., 2020. Describing the trend of ammonia, particulate matter and nitrogen oxides: The role of livestock activities in northern Italy during Covid-19 quarantine. Environmental Research 191, 110048. https://doi.org/10.1016/j.envres.2020.110048

These studies could be commented to discuss NH3 emissions.

The two studies have been included in the revised version of the manuscript (see answer to 2nd specific comment)

Gkatzelis, G.I., Gilman, J.B., Brown, S.S., Eskes, H., Gomes, A.R., Lange, A.C., McDonald, B.C., Peischl, J., Petzold, A., Thompson, C.R., Kiendler-Scharr, A., 2021. The global impacts of COVID- 19 lockdowns on urban air pollution. Elementa: Science of the Anthropocene 9, 00176. https://doi.org/10.1525/elementa.2021.00176

In the introduction, I would cite this study that is a good review on the impact of COVID restrictions on air quality.

Following the reviewer's suggestion, the Gkatzelis et al. (2021) study has been cited in the introduction section of the manuscript as follows:

"Results from these any many other works (more than 200) have been reviewed and summarised by Gkatzelis et al. (2021)."

Technical comments

Very few minor comments, the paper is very well written, just chose either American or UK English. To help the reader I would add a list of acronyms and abbreviations.

A new Appendix has been added with a list of acronyms and abbreviations

Through the publication, please correct Le Quéré with two accents (when necessary).

Corrected

Line 156: heterogeneous impact (same L 839 and L 1005)

Corrected

Line 239: This sentence could be improved "Italy is where the recovery is more pronounced, reaching emissions above the BAU during August"

Sentence has been modified as follows:

"The most pronounced recovery occurs in Italy, where emissions reach levels above BAU during August"

Line 394: I would write "Until the end of August, most"

Corrected

**Anonymous Referee #2**

The manuscript developed a comprehensive European dataset of emission adjustment factors due to COVID-19 at daily basis for each country in 2020. A total of nine sectors is included in the dataset. Combining the emission adjustment factors, as well as the basic pre-COVID emission inventory, the dataset can serve the atmospheric modeling community on the analyses regarding the emission contributions by countries, sectors, pollutants and days during the pandemic. The manuscript is generally clearly organized, thorough discussions are provided, and dataset is publicly available. I just have several technical comments on the data sources in deriving the emission changes. Below are my detailed comments.

Line 17: You didn't mention the methodology and dataset used in this work. Can you summarize the key method and dataset in deriving the emission adjustment factors for key sectors in one sentence or two in the abstract?

The following sentence has been added in the abstract of the revised manuscript:

"We considered metrics traditionally used to estimate emissions, such as energy statistics or traffic counts, as well as information derived from new mobility indicators and machine learning techniques"

Line 33: -51%

Corrected

Line 35-36: I don't think a reference or doi here is appropriate in the abstract. The same for the BAU gridded inventory.

Following the ESSD manuscript composition guidelines: "a functional data set DOI and its in-text citation must be given in the abstract"

https://www.earth-system-science-data.net/submission.html

Line 39: There are 17 figures in the paper... too many for the readers to follow the key analyses. Can you simplify them or moving some of them to supporting information?

Following the reviewer's suggestion, Figures 2, 5 and 8 of the original manuscript have been moved to the supplementary material. The sections of the manuscript referring to these figures have been modified accordingly. The numbering of the figures has been updated accordingly.

Line 192: For the power industry, do you have statistics of the power generation? I understand the power generation can be related to the outdoor temperature (for AC), but there are lots of other electricity needs which are not directly related to temperature (like cooking). It's more straightforward and reliable to use the power production or fuel consumption statistics to derive the emissions.

In this study, we used gradient boosting machine (GBM) models for predicting the fluctuations of electricity demand based on the temperature, assuming that temperature is a strong driver of electricity demand (for heating and air conditioning) and neglecting other factors that can influence demand variability such as change of technology or behaviour (or the fact that some electricity needs like cooking are not directly related to temperature, as the reviewer indicates). The performance of the GMB models, and thus of the assumption that we made, was evaluated in Guevara et al. (2021), where GMB modelled versus measured electricity demand levels were compared for the first two months of 2020, before COVID-19 restrictions entered into force. A high correlation (above 0.9) and

low NMB and NRMSE (below 5 %) were observed for most of the countries, especially in those with stronger lockdown restrictions such as Italy, France or Spain, which give confidence in our approach.

Having said that, it is true that the use of power production or fuel consumption statistics would, in principle, be more reliable to derive emission changes. However, the following limitations arise:

- Electricity generation (or power production): ENTSO-E provides near-real time electricity generation data per production type (i.e., per fuel type) for individual EU countries. However, deriving 2020 business-as-usual electricity generation levels is a much more complex task than when working with electricity demand data, as fluctuations in the electricity generation from fossil fuels does not only depend on the outdoor temperature, but also on other meteorological parameters that influence the renewable power generation capacity (e.g., wind speed and solar radiation) as well as economic factors (i.e., fluctuations in the price of the fuels). In the present study we assumed that changes in the electricity demand levels were affecting electricity generation levels in a homogeneous way across all types of sources (i.e., a drop in energy demand levels implies that both fossil fuel and renewable power plants reduce equally their activity). However, a study by the IEA (2021) suggests that during the first lockdown period changes occurred not only in electricity demand levels but also on the electricity mix. In the case of Europe, results indicate that the power mix slightly shifted towards renewables due to low operating costs and priority access to the grid through regulations, among others. The study also suggests that the electricity mix when back to previous trend with the easing of the restrictions.
  We have added a new point in the Limitations section of the revised manuscript (Section 6.1) to highlight this aspect:
  "For the public power industry sector, we assumed that changes in the electricity demand were affecting electricity generation levels homogeneously across all types of sources (i.e., a drop in energy demand implies that both fossil fuel and renewable power plants reduce equally their activity). However, a study by IEA (2021) suggests that during the first lockdown period changes occurred not only in electricity demand levels but also on the electricity mix. In the case of Europe, results indicate that the power mix slightly shifted towards renewables due to low operating costs and priority access to the grid through regulations, among others. This effect was heterogeneous across countries. The study also suggests that the electricity mix shifted back to the previous trend with the easing of the restrictions."

  *IEA: Covid-19 impact on electricity. Available at: https://www.iea.org/reports/covid-19-impact-on-electricity (last accessed: March 2022), 2021.*

- Fuel consumption statistics in the public power industry sector: To the authors' knowledge, and unlike in the case of electricity demand/generation, this type of information is not reported in an open access, near-real time and consistent way at a daily resolution across Europe. Therefore, it could not be considered in the present study.

Line 337: I'm confused for the usage of Google Mobility Reports in estimating the emissions of "other stationary combustion activities". Only mobility trends can be reflected, so even they appear in places like restaurants, they cannot represent the emissions emitted by the restaurants.

A proper estimation of the impact of COVID-19 restrictions on this sector would require the use of energy proxies that distinguish between what occurred in the residential and commercial/institutional sectors. However, and to the authors' knowledge, there is currently no open access dataset that provides near-real time and high temporal resolution information on European energy consumption

for the residential and commercial sectors separately. Taking into account this lack of information, we made use of the Google mobility data, as it reports changes in daily movement trends across different categories of places, including commercial/institutional places (i.e., groceries and pharmacies, retail and recreation and workplaces) and residential areas. Knowing that changes in movements do not necessarily represent changes in the energy consumption, the original google mobility data was scaled making use of energy consumption statistics available from current literature.

Following with the answer provided to the 1st comment of Anonymous Referee #1, we have highlighted the limitation of available EU energy consumption statistics and of our approach in the revised version of the manuscript as follows:

Adjustment factors for the residential and commercial stationary combustion sectors were derived from Google mobility statistics, which may not necessarily represent changes in the energy consumption of these two sources. However, we could not find any open access dataset that provides near-real time and high temporal resolution information on European energy consumption for the residential and commercial sectors separately. The closest dataset meeting these characteristics is the ENTSO-G transparency platform (https://transparency.entsog.eu/, last accessed March 2022), which reports data on EU daily natural gas flows toward distribution and final consumption. However, the data does not separate commercial/public and residential buildings and it is only available for a limited number of EU countries. There are other national databases that face similar problems, such as GRTgaz (https://www.smart.grtgaz.com/en/consommation/GRTgaz, last accessed March 2022), which provides daily consumption of natural gas by industrial sites and the public network in France, without distinguishing between commercial/institutional and residential sectors."

Line 499, Line 550: How about the gasoline-fueled vehicles?

The emission adjustment factors were computed for Light duty and heavy-duty vehicles (LDV and HDV) and then combined with the GNFR categories reported by the CAMS-REG BAU inventory, which distinguishes between fuels (gasoline, diesel and liquified petroleum gas, LPG). GNFR_F1 includes LDV gasoline-exhaust traffic emissions, GNFR_F21 includes LDV diesel-exhaust traffic emissions, GNFR_F22 reports HDV diesel-exhaust traffic emissions and GNFR_F3 includes LDV LPG-exhaust traffic emissions.

Line 957: Where are the machine learning techniques used? Anything I missed?

Machine learning techniques were used to compute 2020 business-as-usual (BAU) electricity demand levels and subsequently derive emission adjustment factors for the public power industry sector (Section 3.1.1 of the revised manuscript). As described in the manuscript, gradient boosting machine (GBM) models were trained and tuned independently for each country to compute daily BAU electricity demand levels and compare them to the measured electricity statistics reported for 2020 (with the COVID-19 effect). The difference between the daily BAU and measured 2020 electricity demand levels were used to derive country-dependent daily emission adjustment factors.

To clarify this point, the following sentence has been added to the revised version of the manuscript:

"Meteorological data and machine learning techniques were used to compute the differences between measured 2020 electricity demand levels and what would have occurred in the absence of COVID-19."